# Towards end-to-end automation of AI research

Chris Lu[1,2,5], Cong Lu[1,3,4,5], Robert Tjarko Lange[1,5], Yutaro Yamada[1,5✉], Shengran Hu[1,3,4], Jakob Foerster[2], David Ha[1✉] & Jeff Clune[3,4✉]

The automation of science is a long-standing ambition in artificial intelligence (AI) research[1,2]. Although the community has made substantial progress in automating individual components of the scientific process, a system that autonomously navigates the entire research life cycle—from conception to publication—has remained out of reach. Here we present a pipeline for automating the entire scientific process end to end. We present The AI Scientist, which creates research ideas, writes code, runs experiments, plots and analyses data, writes the entire scientific manuscript, and performs its own peer review. Its ideas, execution and presentation are of sufficient quality that the manuscript generated by this AI system passed the first round of peer review for a workshop of a top-tier machine learning conference. The workshop had an acceptance rate of 70%. Our system leverages modern foundation models[3–5] within a complex agentic system. We evaluate The AI Scientist in two settings: a focused mode using human-provided code templates as an initial scaffold for conducting research on a specific topic and a template-free, open-ended mode that leverages agentic search for wider scientific exploration[6,7]. Both settings produce diverse ideas and automatically test, report on and evaluate them. This achievement demonstrates the growing capacity of AI for making scientific contributions and signifies a potential paradigm shift in how research is conducted. As with any impactful new technology, there could be important risks, including taxing overwhelmed review systems and adding noise to the scientific literature. However, if developed responsibly, such autonomous systems could greatly accelerate scientific discovery.

AI has long been used to aid scientific discovery, an ambition with deep roots in the history of the field[1,8–11]. Before the rise of large language models (LLMs), AI was limited to helping with specific, narrow tasks, such as discovering chemical structures[2], finding mathematical proofs[1], discovering new materials[12–14] and predicting the three-dimensional shape of proteins[15,16]. Other systems focused on analysing pre-collected datasets to find new insights[10,17,18]. However, with the recent advent of powerful and general foundation models, the role of AI has expanded to include assisting with a wider array of research activities. For example, LLMs now help with generating new hypotheses[19–23], writing literature reviews[24,25] and coding experiments[26–29]. Despite these advances in automating individual components, a system that autonomously navigates the entire research life cycle—from conception to publication—has remained out of reach until now.

This paper introduces The AI Scientist, a pipeline that achieves the vision of full end-to-end automation of the scientific process. The AI Scientist uses existing foundation models to perform ideation, literature search, experiment planning and implementation, result analysis, manuscript writing, and peer review to produce complete, new papers. We focus on machine learning science, as experiments typically occur entirely on the computer.

A central challenge in developing such a system is automatically evaluating the quality of its scientific output at scale. To address this, we created an automated reviewer and first evaluated its performance against real, human-generated papers. The Automated Reviewer can accurately predict conference acceptance decisions, performing on par with human reviewers (Supplementary Information section A.3). We then used The Automated Reviewer to compare various configurations of The AI Scientist by assessing how performance changes with the scale of the test-time compute and the quality of the underlying foundation model. We find that The AI Scientist performs better with more compute resources (Fig. 3c). Furthermore, The Automated Reviewer shows that improvements to the base models significantly improve the quality of the generated papers, a finding that strongly implies that future versions of our system will be substantially more capable, as models continue to improve (Fig. 1b).

To assess The AI Scientist in the same setting in which human-authored papers are evaluated, we conducted an experiment where we submitted generated papers to a workshop at the International Conference on Learning Representations (ICLR), with the organizers' consent. In computer science, such top-tier conferences are the primary and most prestigious venues for archival and rigorously peer-reviewed

[1]Sakana AI, Tokyo, Japan. [2]FLAIR, University of Oxford, Oxford, UK. [3]University of British Columbia, Vancouver, British Columbia, Canada. [4]Vector Institute, Toronto, Ontario, Canada. [5]These authors contributed equally: Chris Lu, Cong Lu, Robert Tjarko Lange, Yutaro Yamada. ✉e-mail: yutaro.yamada.y@gmail.com; hadavid@sakana.ai; jclune@gmail.com

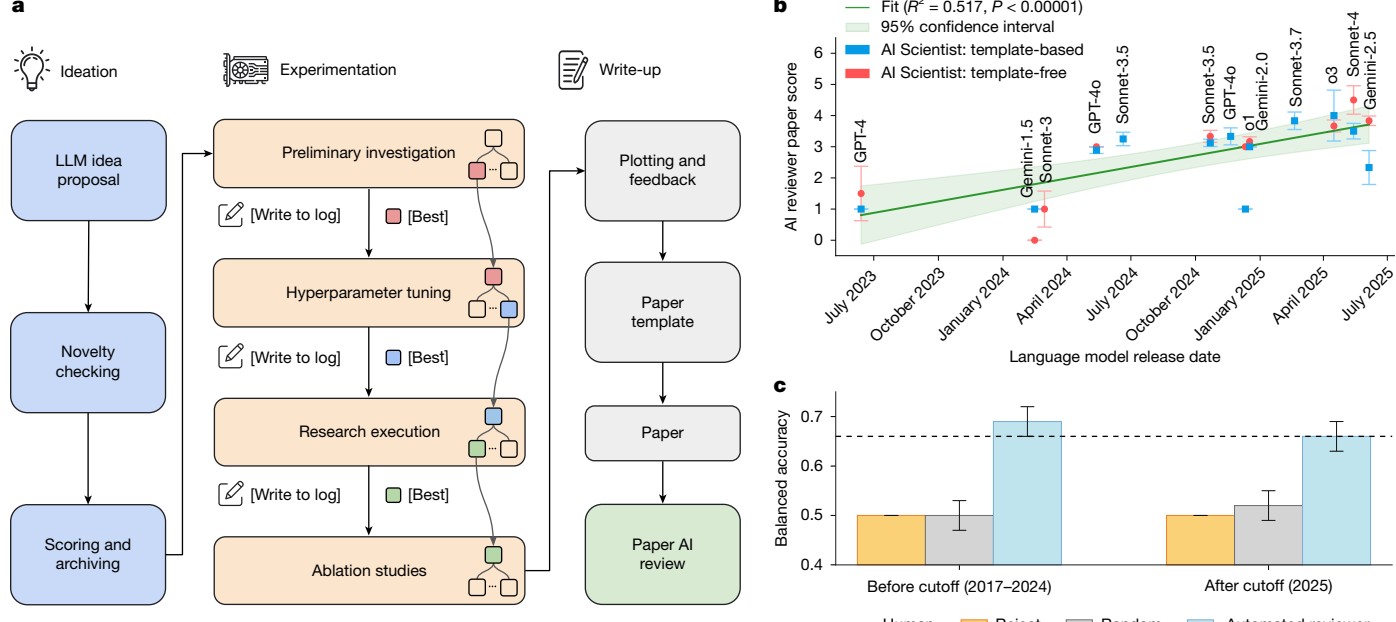

**Fig. 1 | The AI Scientist workflow. a**, The AI Scientist consists of distinct phases covering automated idea generation, tree-based experimentation, manuscript writing and reviewing. The experimentation phase uses an agentic tree search to generate and refine code implementations. This is structured into four stages: (1) initial investigation, (2) hyperparameter tuning, (3) research agenda execution and (4) ablation studies. From one experimental stage to the next, the best-performing checkpoint is selected to seed the next stage of the tree search. **b**, Scores for The AI Scientist papers across model releases. Paper quality consistently improves with the underlying model release date (as judged by The Automated Reviewer), indicating consistent future improvements with improving foundation models. The observed correlation is statistically significant ($P < 0.00001$). Shaded regions represent the standard error. Points represent mean scores with error bars and shaded regions indicating the standard error ($n = 6$ for template-free points, $n = 3$ for template-based points). Full experimental details, including model versions and replication counts, are provided in Supplementary Information section A.2.9. **c**, Automated review versus conference decisions. The Automated Reviewer achieves performance comparable with that of human reviewers, as validated by openly available decisions from past conferences (Table 1). Bars represent mean balanced accuracy; error bars show 95% bootstrapped confidence intervals (5,000 replicates). For replicability, each automated review is a 5-run ensemble. Two-sample $z$-tests on subsampled accuracy (automated $n = 698/876$, human $n = 412$) showed no significant difference before the training cutoff ($P = 0.319$) or post-cutoff ($P = 0.921$). Non-parametric bootstrap tests on $F_1$ scores showed automated outperformance ($P < 0.001$).

publication. They also have workshops with a substantially lower but still non-trivial bar for peer-reviewed acceptance.

One of The AI Scientist's manuscripts achieved high enough scores to exceed the average human acceptance threshold at a workshop, providing an example of a fully AI-generated paper successfully navigating a peer-review process, albeit one with a lower bar.

## Generating manuscripts

The AI Scientist sequentially completes four main phases (Fig. 1a). In the first phase, The AI Scientist is prompted to iteratively grow an archive[30] of high-level research directions and hypotheses that it can explore within a user-specified machine learning research subfield (an example progression is visualized in Supplementary Information section C.4). For each direction, it generates a descriptive title, its reasoning for what the idea is and why it would be interesting to pursue, and a proposed experimental plan (Supplementary Information sections A.1.1 and A.2.6). After idea generation, The AI Scientist filters ideas by connecting the language model to the Semantic Scholar application programming interface (API)[31] and web access as tools[32]. This allows The AI Scientist to discard any idea that too closely resembles a work in the existing literature.

The second phase of The AI Scientist executes the proposed experiments and then visualizes their results for the downstream write-up. We tested two different variants of experiment execution: (1) Template-based: The AI Scientist is provided with a starting code template that reproduces a training run from a popular algorithm. The AI Scientist then executes the proposed experiment plan in linear order (Supplementary Information section A.1). (2) Template-free: Alternatively, The AI Scientist can generate an initial starting code script by itself. In this case, experimentation includes further stages for optimizing the code it writes from scratch, and experiment execution leverages extra test-time compute with a tree search (Fig. 3a,b and Methods). After each experiment, The AI Scientist is given the results and is prompted to take notes in the style of an experimental journal for future planning and write-up.

The third phase of The AI Scientist produces a concise write-up of its research in the style of a standard machine learning conference paper. The AI Scientist is prompted to fill in a blank LaTeX conference template section by section using its notes and plots (Methods). To construct the related work section and add citations throughout the manuscript, the system queries the Semantic Scholar[31] API for relevant literature and compares its findings against the generated manuscript over 20 rounds. For each potential citation, the system generates a textual justification for its inclusion, which informs The AI Scientist on how to use the reference appropriately within the manuscript.

Finally, the paper generated by The AI Scientist undergoes a review by The Automated Reviewer, which automatically evaluates the scientific quality of the conducted research.

## Automated evaluation of generated papers

The Automated Reviewer provides reviews based on the review guidelines for the top-tier Neural Information Processing Systems (NeurIPS) conference (https://neurips.cc/Conferences/2022/

**Table 1 | Performance comparison of human reviewers and The Automated Reviewer**

| Reviewer | Balanced accuracy (↑) | Accuracy (↑) | $F_1$ score (↑) | AUC (↑) | FPR (↓) | FNR (↓) |
|---|---|---|---|---|---|---|
| Human (NeurIPS) | 0.66 | 0.73 | 0.49 | 0.65 | 0.17 | 0.52 |
| Years before knowledge cutoff (2017–2024) | | | | | | |
| Random decision | 0.50 | 0.54 | 0.47 | 0.52 | 0.47 | 0.43 |
| Always reject | 0.50 | 0.65 | 0.00 | 0.50 | 0.00 | 1.00 |
| Automated Reviewer | 0.69 ± 0.04 | 0.65 ± 0.10 | 0.62 ± 0.09 | 0.69 ± 0.09 | 0.45 ± 0.10 | 0.17 ± 0.08 |
| Year after knowledge cutoff (2025) | | | | | | |
| Random decision | 0.52 | 0.51 | 0.48 | 0.49 | 0.50 | 0.48 |
| Always reject | 0.50 | 0.56 | 0.00 | 0.50 | 0.00 | 1.00 |
| Automated Reviewer | 0.66 ± 0.03 | 0.63 ± 0.09 | 0.67 ± 0.09 | 0.65 ± 0.10 | 0.52 ± 0.10 | 0.17 ± 0.07 |

Performance comparison of human reviewers (NeurIPS 2021 consistency experiment[34]) and the Automated Reviewer, evaluated on papers published before (2017–2024) and after (2025) the knowledge cutoff. The Automated Reviewer achieved performance superior or comparable with human reviewer consistency in key metrics such as $F_1$ score, area under the curve (AUC) and balanced accuracy, even for data beyond the knowledge cutoff, highlighting its robustness and reliability across different time periods. Error margins denote the 95% bootstrapped confidence intervals. Arrows indicate whether it is better for a score to be higher (↑) or lower (↓). Supplementary Information section A.3.2 explains each metric and comparison in detail. FNR, false negative rate; FPR, false positive rate.

Reviewer Guidelines). The output contains numerical scores (soundness, presentation, contribution, overall quality and reviewer confidence), lists of weaknesses and strengths, as well as a binary decision (accept or reject). The Automated Reviewer pipeline consists of an ensemble of five reviews, followed by a meta-review in which the model acts as an area chair to make a final decision conditioned on all five reviews (Supplementary Information section A.3). We compared Automated Reviewer decisions with ground truth data for ICLR papers extracted from the publicly available OpenReview dataset[33]. As shown in Table 1, the agreement of Automated Reviewer assessments with human assessments is comparable with inter-human agreement measured by $F_1$ score and balanced accuracy, as reported in the NeurIPS 2021 consistency study[34], which measured agreement between human reviewers on a comparable set of submissions (Supplementary Information section A.3). This demonstrates its ability to replicate the collective judgement of human reviewers with high fidelity. These results are statistically significant (non-parametric bootstrap test[35] and two-sample $z$-test[36]; Supplementary Information section A.3). Next, to investigate the effect of potential data contamination (the possibility that decisions on a paper were part of the training set for the LLM), we evaluated The Automated Reviewer on two datasets: one containing 1,000 papers from years potentially within the training data used for the model (2017–2024) and a second 'clean' dataset from the year after the cutoff (2025), which could not have been seen during training. A comparison between years before and after the knowledge cutoff indicates that data contamination may exist, as balanced decision accuracy decreases from 69% before to 66% in the year after the cutoff. However, the results for the year after the cutoff remain comparable with those of human reviewers (for example, 66% balanced accuracy), showing that potential contamination had, at most, a minimal effect.

Using The Automated Reviewer, we assessed the quality of the research papers generated by a wide range of LLMs as the core model within The AI Scientist. Our analysis revealed a clear trend: as models improve over time, the quality of the papers produced by The AI Scientist increased correspondingly (Fig. 1b). With recent generations of models, on average, The AI Scientist produced papers that approach borderline acceptability for machine learning conference workshops, as judged by our Automated Reviewer (Supplementary Fig. B2). Additionally, there is a strong correlation between the amount of compute allocated per paper and the resulting quality (Fig. 3c), indicating that both model scale and inference-time investment play important roles in the output quality of The AI Scientist, further indicating the possibility of substantial improvements as the costs of AI systems continue to exponentially decrease and capabilities exponentially increase[37].

## Human evaluation results

Perhaps the ultimate and fairest test of the quality of the work of The AI Scientist is a version of what we might call an AI scientist Turing test: submitting the work to the same rigorous, blind peer-review systems used to evaluate human science. We submitted three generated manuscripts to the formal peer-review process of a workshop at a top-tier machine learning conference. This experiment was conducted with the approval of the relevant institutional review board (IRB; Supplementary Information section C.3) and the full cooperation of the ICLR 2025 leadership and the organizers of the I Can't Believe It's Not Better (ICBINB) workshop. This was the only venue that we submitted to.

The template-free version of The AI Scientist was readily adapted to this setting by simply prompting it with the broad theme of the workshop (which was investigating deep learning limitations, including where previous ideas to improve it had not worked). The overall process was then run to generate ideas, experiments and papers. We manually filtered the most promising outputs at each stage (Supplementary Information section A.4). Had this filtering not occurred, the papers under analysis would still have been produced in their final form, just along with other papers and, thus, at a greater total cost. This process resulted in three complete manuscripts being selected for submission. The selection was based on three criteria: whether the idea was aligned with the workshop topic, whether the code correctly implemented the proposed idea and ran without errors, and the correctness of the manuscript formatting (Supplementary Information section A.4). The entire scientific workflow for each paper, from ideation and coding to manuscript writing, was performed without any human modification. These three submissions were included among the 43 papers reviewed for the workshop. Reviewers were informed that some of the submissions were AI-generated but not which ones, ensuring a blind process.

One of the three AI-generated manuscripts received an average score of 6.33 from the reviewers (individual scores were 6, 7 and 6), placing it above the average acceptance threshold for the workshop (Fig. 2). The organizers said that the paper would have been accepted in all likelihood were it not withdrawn according to our pre-established protocol due to being AI-generated. Notably, the accepted manuscript reported a negative result, aligning with the focus of the workshop on interesting negative results. The other two papers did not meet the bar for acceptance (Supplementary Table D9). Thus, a fully AI-generated paper passed a standard scientific peer-review process. We also conducted our own internal review, using the human AI researchers on our team (Supplementary Information section C.2). The team concluded that although one of the papers did meet the bar for workshop papers, none met the higher bar for a main ICLR conference publication. A

## Compositional Regularization: Unexpected Obstacles in Enhancing Neural Network Generalization

**AI Scientist-v2**
Paper under double-blind review

### Abstract

Neural networks excel in many tasks but often struggle with compositional generalization—the ability to understand and generate novel combinations of familiar components. This limitation hampers their performance on tasks requiring systematic reasoning beyond the training data. In this work, we introduce a training method that incorporates an explicit compositional regularization term into the loss function, aiming to encourage the network to develop compositional representations. Contrary to our expectations, our experiments on synthetic arithmetic expression datasets reveal that models trained with compositional regularization do not achieve significant improvements in generalization to unseen combinations

**Title and abstract (page 1)**

Our goal is to enhance compositional generalization in neural networks by incorporating a compositional regularization term into the training loss. We focus on a simple yet illustrative task: evaluating arithmetic expressions involving basic operators.

#### 3.1 Model Architecture

We use an LSTM-based neural network (Goodfellow et al., 2016) to model the mapping from input expressions to their computed results. The model consists of an embedding layer, an LSTM layer, and a fully connected output layer.

#### 3.2 Compositional Regularization

Let $h_t$ be the hidden state at time $t$. We define the compositional regularization term as the mean squared difference between successive hidden states:

$$L_{\text{comp}} = \frac{1}{T-1} \sum_{t=1}^{T-1} \|h_{t+1} - h_t\|^2 \qquad (1)$$

where $T$ is the length of the input sequence.

This term penalizes large changes in hidden states between successive time steps, encouraging the model to form additive representations, which are a simple form of compositionality.

#### 3.3 Training Objective

**Technical methodology (page 2)**

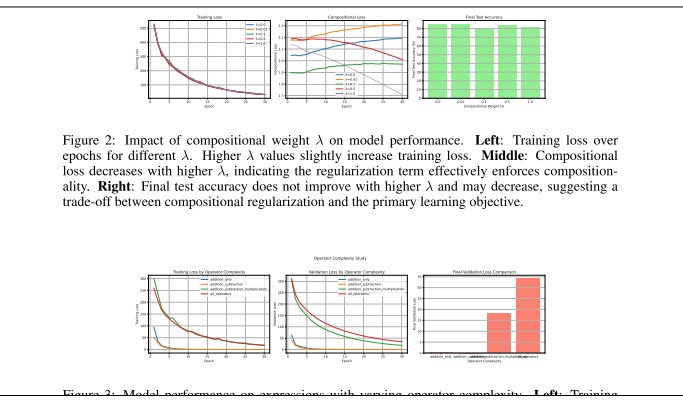

Figure 2: Impact of compositional weight $\lambda$ on model performance. **Left**: Training loss over epochs for different $\lambda$. Higher $\lambda$ values slightly increase training loss. **Middle**: Compositional loss decreases with higher $\lambda$, indicating the regularization term effectively enforces compositionality. **Right**: Final test accuracy does not improve with higher $\lambda$ and may decrease, suggesting a trade-off between compositional regularization and the primary learning objective.

Figure 3: Model performance on expressions with varying operator complexity. **Left**: Training

**Data visualizations (page 4)**

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

**References (page 5)**

**Fig. 2 | Selected sections from a paper generated by The AI Scientist that was accepted via peer review at a top-tier machine learning conference workshop.** The paper received peer-review scores of 6 (weak accept), 7 (accept) and 6 (weak accept) before meta-review and ranked among the top 45% of papers submitted for peer review. This demonstrates that a fully AI-generated paper can navigate the peer-review process successfully at a top-tier conference workshop. A full-sized version of this paper is available in Supplementary Information section D.2.1.

full analysis of all three submitted papers, including their strengths, weaknesses and implementations, is provided in Supplementary Information section C.2.

## Limitations

Although The AI Scientist generated a workshop paper that passed peer review, there is room for improvement if it is to match the best human-produced science. Only one of three submissions was accepted, and workshops have much higher acceptance rates than main conferences (for example, 70% for the ICLR 2025 ICBINB workshop[38] versus 32% for the ICLR 2025 main conference[39]). Therefore, The AI Scientist cannot yet meet the standards of top-tier publications nor even do so consistently for workshops. Common failure modes include the generation of naive or underdeveloped ideas, incorrect implementations of the main idea, a lack of deep methodological rigour, errors in experimental implementation, duplicating figures in the main text and the appendix, and many types of hallucinations, such as inaccurate citations (a full analysis of failure modes is provided in Supplementary Information sections A.4, C.2 and C.3).

That said, often in machine learning, once something begins to work (even with clear flaws), in a few short years with scale (for example, of compute and data), better core models and better techniques, the capabilities of a system become surprising and can exceed human performance levels. In assessing the impact of a technology, it is, thus, important to keep in mind its probable future trajectory. Crucially, this trajectory is not just about better models but about the complexity of the tasks that AI systems can execute. Recent work indicates that the length of tasks that AI can reliably complete is doubling every 7 months[40], indicating that many current implementation and debugging bottlenecks may be resolved in the near term. However, some AI weaknesses have proved surprisingly difficult to solve, such as AI being easily fooled[41,42] and overconfidently wrong (hallucinations)[43], although progress has been made[44,45]. Such challenges could persist and would prevent us from reliably trusting the outputs of systems like The AI Scientist. It is also not clear to what extent AI systems can produce new creative ideas that resemble great conceptual leaps in science. Studying and improving AI systems on these fronts are key areas for future research.

At present, The AI Scientist conducts computational experiments only. In future work, this same playbook could be applied to other scientific domains where one can automatically conduct experiments (or have humans conduct them) and collect data from them (for example, automated chemistry laboratories, on which swift progress is being made[46]).

The ability to automate paper generation raises important ethical and societal concerns, including the potential to overwhelm the peer-review process, artificially inflate research credentials, repurpose the ideas of others without giving proper credit, eliminate scientist jobs, or conduct unethical or dangerous experiments (Supplementary Information section C.3). To conduct this study responsibly, we obtained explicit permission from the ICLR leadership, the workshop organizers and the University of British Columbia's IRB (H24-02652). Crucially, as part of our experimental protocol, we determined in advance that all AI-generated submissions would be withdrawn after peer review, regardless of outcome. This decision was made to avoid setting a precedent for publishing fully automated research before the scientific community has established clear standards for disclosure and

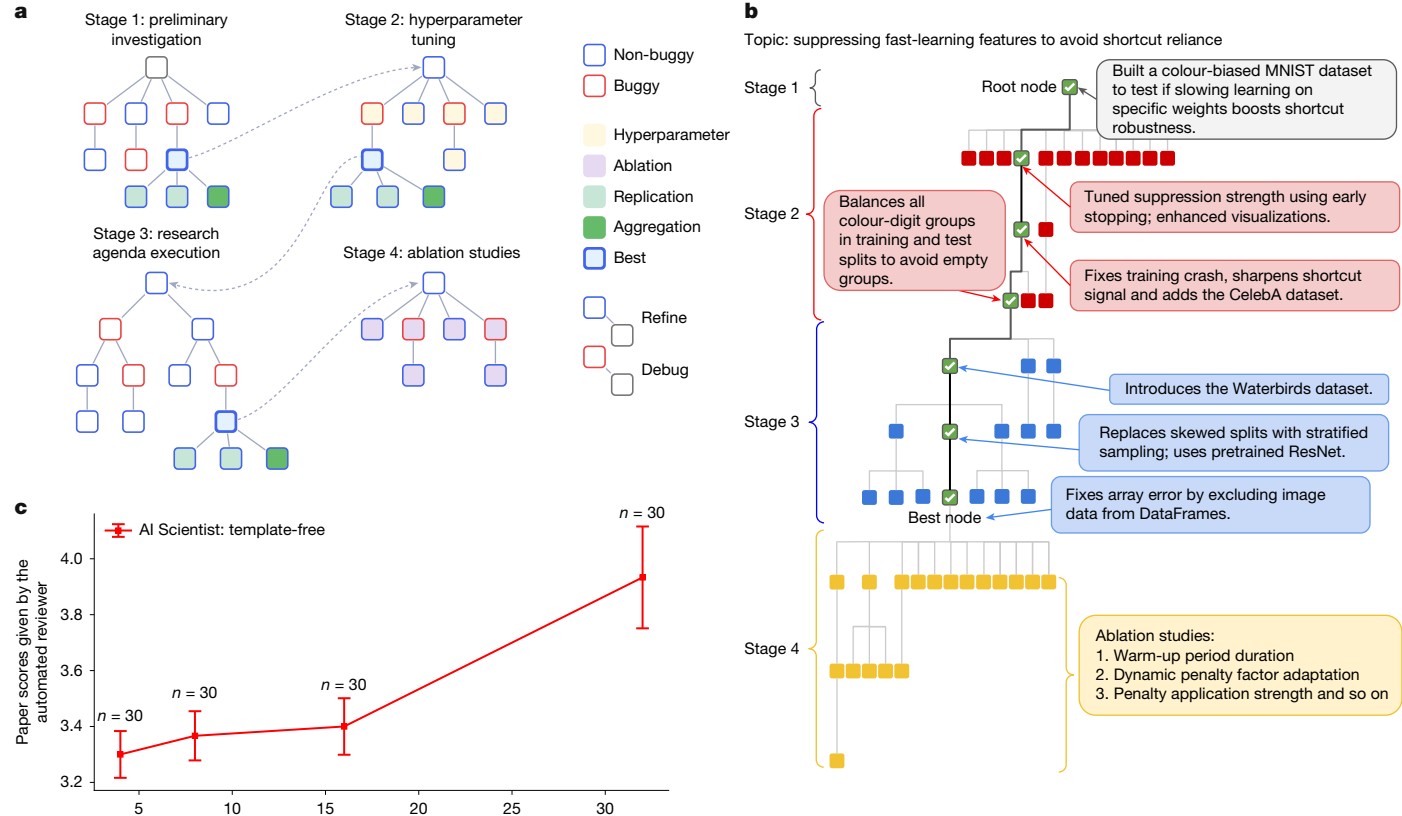

**Fig. 3 | The phases and compute scaling of the AI Scientist. a**, The research experimentation phase is visualized as a four-stage process. A preliminary baseline code implementation is first constructed (stage 1) and refined by tuning the hyperparameters (stage 2). The resultant code serves as a starting point for executing the research agenda through an agentic tree search (stage 3), followed by ablation experiments (stage 4). Full details of the agentic tree search process are provided in Methods. **b**, A real example of tree search by The AI Scientist with node annotations outlining the experiments conducted in the four different stages. **c**, Scores for The AI Scientist papers across compute budgets. Scaling the number of experimental nodes in the agentic tree search shows improvements for deeper test-time search budgets. Error bars represent the standard error. Full experimental details are provided in Supplementary Information section A.2.9.

evaluation. Developing these norms is a critical next step to ensure that such systems are used to advance, not undermine, scientific integrity. Finally, more research is needed to ensure that open-ended exploratory AI proceeds safely and in alignment with human values[47,48].

The generation by The AI Scientist of an AI-authored manuscript that passed peer review for a workshop at a top-tier machine learning conference marks a milestone in the centuries-long scientific endeavour. Although challenges remain in terms of consistency and achieving top-tier quality, this success demonstrates the growing capacity of AI for scientific reasoning, and it signals the dawn of a new era in which the process of discovery is no longer a solely human pursuit and in which the pace at which we are able to reap the harvest of scientific discovery could accelerate dramatically.

## Online content

1. Lenat, D. B. Automated theory formation in mathematics. In *Proc. 5th International Joint Conference on Artificial Intelligence* 833–842 (ed. Reddy, R.) (William Kaufmann, 1977).
2. Buchanan, B. G. & Feigenbaum, E. A. Dendral and meta-dendral: their applications dimension. *Artif. Intell.* **11**, 5–24 (1978).
3. OpenAI. GPT-4 technical report. Preprint at https://doi.org/10.48550/arXiv.2303.08774 (2023).
4. *The Claude 3 Model Family: Opus, Sonnet, Haiku* https://api.semanticscholar.org/CorpusID:268232499 (Anthropic, 2024).
5. Grattafiori, A. et al. The Llama 3 herd of models. Preprint at https://doi.org/10.48550/arXiv.2407.21783 (2024).
6. Jiang, Z. et al. AIDE: AI-driven exploration in the space of code. Preprint at https://doi.org/10.48550/arXiv.2502.13138 (2025).
7. Chan, J. S. et al. MLE-bench: evaluating machine learning agents on machine learning engineering. In *Proc. 13th International Conference on Learning Representation* (ed. Yue, Y. et al.) (ICLR, 2025).
8. Waltz, D. & Buchanan, B. G. Automating science. *Science* **324**, 43–44 (2009).
9. Langley, P. Integrated systems for computational scientific discovery. In *Proc. AAAI Conference on Artifical Intelligence* Vol. 38, 22598–22606 (eds Wooldridge, M. et al.) (AAAI, 2024).
10. Langley, P., Simon, H. A., Bradshaw, G. L. & Zytkow, J. M. *Scientific Discovery: Computational Explorations of the Creative Process* (MIT, 1987).
11. Lenat, D. B. & Brown, J. S. Why AM and EURISKO appear to work. *Artif. Intell.* **23**, 269–294 (1984).
12. Pyzer-Knapp, E. O. et al. Accelerating materials discovery using artificial intelligence, high performance computing and robotics. *npj Comput. Mater.* **8**, 84 (2022).
13. Merchant, A. et al. Scaling deep learning for materials discovery. *Nature* **624**, 80–85 (2023).
14. Szymanski, N. J. et al. An autonomous laboratory for the accelerated synthesis of novel materials. *Nature* **624**, 86–91 (2023).
15. Jumper, J. et al. Highly accurate protein structure prediction with AlphaFold. *Nature* **596**, 583–589 (2021).
16. Hayes, T. et al. Simulating 500 million years of evolution with a language model. *Science* **387**, 850–858 (2025).
17. Ifargan, T., Hafner, L., Kern, M., Alcalay, O. & Kishony, R. Autonomous LLM-driven research—from data to human-verifiable research papers. *NEJM AI* **2**, AIoa2400555 (2025).
18. Falkenhainer, B. C. & Michalski, R. S. Integrating quantitative and qualitative discovery: the ABACUS system. *Mach. Learn.* **1**, 367–401 (1986).
19. Girotra, K., Meincke, L., Terwiesch, C. & Ulrich, K. T. *Ideas Are Dimes a Dozen: Large Language Models for Idea Generation in Innovation*. Working paper (Cornell Univ., 2023).
20. Lehman, J. et al. in *Handbook of Evolutionary Machine Learning* (eds Banzhaf, W. et al.) 331–366 (Springer Nature, 2023).

21. Lu, C., Hu, S. & Clune, J. Automated capability discovery via model self-exploration. Preprint at https://doi.org/10.48550/arXiv.2502.07577 (2025).
22. Faldor, M., Zhang, J., Cully, A. & Clune, J. OMNI-EPIC: open-endedness via models of human notions of interestingness with environments programmed in code. In *Proc. 13th International Conference on Learning Representations* (eds Yue, Y. et al.) (ICLR, 2025).
23. Hu, S., Lu, C. & Clune, J. Automated design of agentic systems. In *Proc. 13th International Conference on Learning Representations* (eds Yue, Y. et al.) (ICLR, 2025).
24. Baek, J., Jauhar, S. K., Cucerzan, S. & Hwang, S. J. ResearchAgent: Iterative research idea generation over scientific literature with large language models. In *Proc. 2025 Conference of the Nations of the Americas Chapter of the Association for Computational Linguistics: Human Language Technologies* (eds Chiruzzo, L. et al.) 6709–6738 (ACL, 2025).
25. Wang, Y. et al. Autosurvey: large language models can automatically write surveys. In *Proc. Advances in Neural Information Processing Systems* Vol. 37 (eds Globerson, A. et al.) 115119–115145 (Curran Associates, 2024).
26. Huang, Q., Vora, J., Liang, P. & Leskovec, J. MLAgentBench: evaluating language agents on machine learning experimentation. In *Proc. 41st International Conference on Machine Learning* Vol. 235 (eds Salakhutdinov, R. et al.) 20271–20309 (PMLR, 2024).
27. Lu, C. et al. Discovering preference optimization algorithms with and for large language models. In *Proc. Advances in Neural Information Processing Systems* Vol. 37 (eds Globerson, A. et al.) 86528–86573 (Curran Associates, 2024).
28. Ma, Y. J. et al. Eureka: human-level reward design via coding large language models. In *Proc. 12th International Conference on Learning Representations* (eds Kim, B. et al.) (ICLR, 2024).
29. Zhang, J., Hu, S., Lu, C., Lange, R. & Clune, J. Darwin Gödel machine: open-ended evolution of self-improving agents. In *Proc. 14th International. Conference Learning Representation* (ed. Vondrick, C. et al.) (ICLR, 2026).
30. Mouret, J.-B. & Clune, J. Illuminating search spaces by mapping elites. Preprint at https://doi.org/10.48550/arXiv.1504.04909 (2015).
31. Fricke, S. Semantic Scholar. *J. Med. Libr. Assoc.* **106**, 145 (2018).
32. Schick, T. et al. Toolformer: language models can teach themselves to use tools. In *Proc. Advances in Neural Information Processing Systems* Vol. 36 (eds Oh, A. et al.) 68539–68551 (Curran Associates, 2023).
33. González-Márquez, R. & Kobak, D. Learning representations of learning representations. In *Proc. Data-centric Machine Learning Research (DMLR) Workshop at ICLR* (ICLR, 2024).
34. Beygelzimer, A., Dauphin, Y., Liang, P. & Vaughan, J. W. The NeurIPS 2021 consistency experiment. *NeurIPS Blog* https://blog.neurips.cc/2021/12/08/the-neurips-2021-consistency-experiment (2021).
35. Efron, B. & Tibshirani, R. J. *An Introduction to the Bootstrap* (Chapman & Hall CRC, 1993).
36. Lehmann, E. L. *Testing Statistical Hypotheses* 1st edn (Wiley, 1959).
37. Maslej, N. et al. Artificial intelligence index report 2024. Preprint at https://doi.org/10.48550/arXiv.2405.19522 (2024).
38. *I Can't Believe it's Not Better: Challenges in Applied Deep Learning. ICLR Workshop at ICLR 2025* http://openreview.net/group?id=ICLR.cc/2025/Workshop/ICBINB (OpenReview, 2025).
39. *13th Annual International Conference on Learning Representations (ICLR) 2025 Fact Sheet* https://media.iclr.cc/Conferences/ICLR2025/ICLR2025_Fact_Sheet.pdf (ICLR, 2025).
40. Kwa, T. et al. Measuring AI ability to complete long tasks. *METR Blog* https://metr.org/blog/2025-03-19-measuring-ai-ability-to-complete-long-tasks/ (2025).
41. Nguyen, A., Yosinski, J. & Clune, J. Deep neural networks are easily fooled: high confidence predictions for unrecognizable images. In *Proc. IEEE Conference on Computer Vision and Pattern Recognition* (eds Grauman, K. et al.) 427–436 (IEEE, 2015).
42. Szegedy, C. et al. Intriguing properties of neural networks. In *Proc. 2nd International Conference on Learning Representations* (eds Bengio, Y. & LeCun, Y.) (ICLR, 2014).
43. Maynez, J., Narayan, S., Bohnet, B. & McDonald, R. On faithfulness and factuality in abstractive summarization. In *Proc. 58th Annual Meeting of the Association for Computational Linguistics* (eds Jurafsky, D. et al.) 1906–1919 (ACL, 2020).
44. *GPT-5 System Card* https://openai.com/index/gpt-5-system-card/ (OpenAI, 2025).
45. Huang, L. et al. A survey on hallucination in large language models: principles, taxonomy, challenges, and open questions. *ACM Trans. Inf. Syst.* **43**, 42 (2025).
46. Boiko, D. A., MacKnight, R., Kline, B. & Gomes, G. Autonomous chemical research with large language models. *Nature* **624**, 570–578 (2023).
47. Bommasani, R. et al. On the opportunities and risks of foundation models. Preprint at https://doi.org/10.48550/arXiv.2108.07258 (2021).
48. Ecoffet, A., Clune, J. & Lehman, J. Open questions in creating safe open-ended AI: tensions between control and creativity. In *Proc. Artificial Life Conference* Vol. 32 (eds Bongard, J. et al.) 27–35 (MIT Press, 2020).

## Methods

Our research methodology is centred around two core automated systems: an AI scientist for generating new scientific research and an automated reviewer for rigorous evaluation. These systems work in concert to explore the potential of AI in accelerating scientific discovery.

### The AI Scientist

The AI Scientist is an agentic system designed to autonomously conduct machine learning research. We present results for two modes: a template-based system that extends human-provided code and a more open-ended template-free system that operates with much less prior guidance. The detailed prompts used for each system are provided in Supplementary Information sections A.1.1 and A.2.6. More results and analyses of the papers generated by each system are provided in Supplementary Information sections B.1, C.1, C.2, D.1 and D.2.

**Foundational technologies.** Both versions are built upon autoregressive LLMs[3–5], which learn to generate text by modelling the conditional probability of a new token given preceding tokens. Through vast data and model scaling, LLMs exhibit human-like abilities, including reasoning and code generation. Agentic patterns[49], such as few-shot prompting[50] and self-reflection[51], are leveraged by The AI Scientist to improve performance and reliability. For code generation, the template-based system uses the state-of-the-art open-source coding assistant Aider[52], which is designed to implement features, fix bugs or refactor code in existing codebases. To go further and effectively use more test-time compute, the template-free system uses LLMs to power a tree search without relying on Aider.

**Template-based AI Scientist.** The system is provided with a starting code template that reproduces a simple training run from a popular algorithm on a standard benchmark (for example, training a small transformer[53] on the works of Shakespeare). Its workflow unfolds in three phases:

1. Idea generation: The process begins with a simple experiment defined by a human-provided code template. The system then enters an iterative loop of idea generation and refinement using LLMs as a mutation operator. In each iteration, it proposes a batch of new research ideas that are variations or extensions of existing ideas in its growing archive. Each idea is a structured object containing a descriptive title, a summary of the core hypothesis, a detailed experimental plan, and self-assessed scores for interestingness (1–10 scale), novelty (1–10 scale) and feasibility (1–10 scale). This iterative growth of an idea archive was inspired by open-endedness algorithms that maintain a diverse collection of artefacts[20,54]. To enforce novelty, each proposed idea is automatically checked against the scientific literature through the Semantic Scholar API[31]; ideas with high semantic similarity to existing works are discarded. The system is prompted to act as an 'ambitious AI PhD student who is looking to publish a paper that will contribute significantly to the field'. For the novelty assessment, the system conducts up to ten rounds of literature search queries, and in each round, the system can refine its search based on previous results.

2. Experiment execution: Once a promising idea is selected from the archive, the system devises a multi-step experimental plan with up to five experiments. It then executes this plan sequentially using Aider to modify the codebase. A key feature of this phase is its robustness to runtime errors. The system automatically detects execution failures, captures the error logs and invokes an instance of the Aider agent[52] to perform automated debugging. The Aider agent is prompted with the failing code and the error message, and it then generates a patch, with up to four reattempt cycles per experiment. The corrected code is then used to rerun the experiment with a timeout of 7,200 s per experiment. All experimental outcomes, including metrics, generated plots and observations, are logged in an experimental journal. This journal serves as a form of memory and informs the subsequent steps in the experimental plan.

3. Manuscript generation: Upon completing the experimental phase, the system synthesizes the findings into a full scientific paper. To do so, it uses Aider to populate a standard conference LaTeX template. Aider writes sections, including the introduction, methods, results and conclusion. The results section is written by analysing the experimental journal, summarizing key findings and embedding the generated figures. To situate the work within the broader scientific context, the system constructs a related work section by querying the Semantic Scholar API for relevant literature (up to 20 search rounds) and generating summaries for each cited paper. The manuscript undergoes several passes of automated editing and refinement to improve clarity and coherence. Finally, the system compiles the LaTeX source and automatically corrects any compilation errors (up to five correction rounds) to produce a final PDF.

**Template-free AI Scientist.** To overcome the limitations of a fixed starting codebase, we developed a template-free version capable of more open-ended discovery. We use OpenAI's o3 for idea generation and code critique during experiments due to its strong reasoning capabilities, Anthropic's Claude Sonnet 4 for code generation, OpenAI's GPT-4o for cost-efficient vision-language tasks and OpenAI's o4-mini for cost-efficient reasoning during the review stage. This version introduces several key enhancements.

**Generalized idea generation.** The ideation process used by the system is more abstract and not tethered to an initial code implementation. It begins by generating high-level research proposals that resemble the abstract of a scientific paper. These proposals articulate a research problem, propose a new method and hypothesize the expected outcomes. To ensure the proposals are both grounded and new, this process is tightly integrated with a literature review module that queries external academic databases to identify knowledge gaps and avoid rediscovering existing work. The system uses structured prompts to guide idea generation and reflection rounds to refine proposals based on the literature search results (see Supplementary Information section A.2.6 for prompts).

**Experiment progress manager.** Real-world scientific experimentation typically proceeds through distinct stages, from initial feasibility assessments to detailed ablation analyses. To emulate this structured approach, we introduced an experiment progress manager to coordinate four clearly defined stages of scientific experimentation: (1) start with a preliminary investigation to test basic viability, (2) tune the hyperparameters for optimization, (3) execute the main research agenda and (4) conclude with ablation studies to understand the contribution of different components. Each stage has explicit stopping criteria. Stage 1 concludes when a basic working prototype has successfully executed. Stage 2 ends when the experiments stabilize, as indicated by convergence in training curves and successful execution across at least two datasets. Stages 3 and 4 conclude when the allocated computational budget is exhausted. Each stage conducts its own tree search. The specifics of this tree search process are detailed in the following bullet point. Each node has a maximum experiment runtime of 1 h. At the end of each stage, an LLM-based evaluator assesses all leaf nodes and selects the most promising one to serve as the root for the next stage of exploration, thus effectively pruning less promising research avenues.

**Parallelized agentic tree search for experimentation.** To manage the complexity of open-ended research, the sequential workflow of the template-based version of The AI Scientist is replaced with a parallelized agentic tree. Figure 3a is an overview of the approach and Fig. 3b shows a tree generated by an actual run. By default, it uses Claude Sonnet 4 for code generation. We provide results for different LLM model choices in Fig. 1b.

Each experimental node within the agentic tree search undergoes the following execution cycle. First, Claude Sonnet 4 generates both a concrete experimentation plan and the associated Python code to implement the experiment. The generated code is immediately executed in a Python interpreter. If the execution encounters an error, the error message is recorded and the node is marked as buggy, ending the current execution cycle for that node. If the execution succeeds, the experiment proceeds to the plotting phase.

The system is prompted to save all relevant experimental outputs (training and validation metrics, losses and so on) into structured numpy files. In the plotting phase, The AI Scientist reads these stored results and the code and generates visualizations that summarize and illustrate the findings. These visualizations are subsequently passed to a vision-language model (VLM) for critique. Any issues flagged by the VLM (such as unclear labels, missing legends or misleading visualizations) result in the node being marked as buggy, and this feedback is recorded for future debugging. Nodes that successfully execute and pass the VLM review without issue are designated as non-buggy.

Each node is defined as a collection comprising an experiment script (for example, a Python file), a textual description of the high-level plan implemented in the script, an execution error trace (if applicable), experiment runtime, performance metrics recorded during the experiment, code critique from OpenAI o3 after running the script, a visualization script, file paths to the generated figures, feedback from a VLM on those figures and the final status of the node (either buggy or non-buggy).

At each iteration, the system selects several nodes from the existing tree to expand in parallel. With a predefined probability, a buggy node is chosen (thus prioritizing error resolution and debugging); otherwise, a non-buggy node is selected for further refinement and improvement. When choosing between non-buggy nodes, the system uses a best-first search strategy guided by GPT-4o, which evaluates candidates based on factors like performance metrics, training dynamics and the quality of the plots generated. The selected nodes are expanded by creating a new child node. The system attempts debugging if the parent node was buggy or refines and improves upon the previous experiment if the parent was non-buggy. Claude Sonnet 4 is used to generate the plan and experiment code for each new child node, after which all new nodes are executed concurrently in parallel, which greatly accelerates the exploration process. In addition to buggy and non-buggy nodes, the system uses specialized node variants tailored to specific experimental needs:

- Hyperparameter nodes systematically explore alternative hyperparameter configurations during stage 2. The system maintains records of previously tested hyperparameters to prevent redundant experiments. Errors encountered during hyperparameter tuning trigger the creation of corresponding debug nodes.
- Ablation nodes evaluate crucial ablation studies during stage 4. This assesses the importance of various components or assumptions underlying the experiment. Like hyperparameter nodes, previously tested ablation conditions are tracked to avoid repetition, and debugging nodes are created in response to any errors encountered.
- Replication nodes execute replicates of their parent experiments using different random seeds. Typically, several replication nodes are created to enable the calculation of statistical measures (mean and s.d.) of experimental outcomes, which enhances the robustness of the results.
- Aggregation nodes are special nodes created to consolidate and visualize the combined results of replication nodes. Unlike other node types, aggregation nodes do not conduct new experiments but simply generate a Python script to aggregate and summarize previous results. The script produces figures that explicitly show mean and s.d.

The structured design of the experimental stages and tailored node types facilitates systematic exploration across all stages. Unlike some LLM agents that rigidly follow predefined, fine-grained workflow graphs, The AI Scientist adopts a looser structure that guides the entire empirical research cycle, thus enabling flexible system behaviour while maintaining coherence across iterative stages. See Supplementary Information sections A.2.6 and A.2.9 for the prompts and detailed hyperparameters, respectively.

**VLM integration.** This system incorporates VLMs using GPT-4o to analyse and provide feedback on visual data. During experimentation, the plots generated are fed to a VLM, which is prompted to act as a scientist and critique them. For example, it might flag nonsensical axes or issues in the quality of generated examples or suggest clearer ways to present the data. This feedback is used to generate new experimental nodes in the tree search aimed at addressing the identified issues. During manuscript preparation, the VLM assesses the alignment between figures and their corresponding captions to ensure that a caption accurately describes the plot and highlights the key takeaways, thus improving the overall quality and clarity of the paper. The VLM reviews include detailed analyses of figure content, caption accuracy and integration with the main text (see Supplementary Information section A.2.6 for prompts).

**Generalized dataset access.** To broaden its research capabilities, the system is prompted to dynamically integrate datasets from public repositories by formulating queries to the HuggingFace Hub[55]. A set of ten example datasets available on HuggingFace is listed in the prompt, and the system can automatically generate the data-loading code needed to use a selected dataset in its experiments. This approach partially relaxes the constraint of working with a fixed, predefined set of datasets by allowing human scientists to easily update the candidate list. For datasets not available on HuggingFace, human scientists can download them from public data repositories (for example, open-access archives), store them locally, and add usage instructions to the prompt. These locally stored datasets can then be used alongside HuggingFace datasets by The AI Scientist (see Supplementary Information section A.2.6 for prompts).

**Enhanced manuscript writing.** The template-free system moves away from the incremental Aider-based approach to direct LaTeX generation using a reasoning model such as OpenAI's o1[56] followed by reflection[51]. The system first aggregates experimental results from several stages into compound figures using a dedicated plot-aggregation step. The manuscript-writing process includes specific prompts for different workshop formats (for example, the ICBINB workshop focusing on negative results), with detailed guidelines for each section, including the title, abstract, introduction, methods, experiments and conclusions. The system undergoes several reflection cycles, each time incorporating feedback from LaTeX linters and VLM reviews to improve figure quality and text–figure alignment (see our code and Supplementary Information section A.2.6 for prompts and full details).

The complete generation process for the template-free system typically takes from several hours to over 15 h, depending on problem complexity.

## The Automated Reviewer

To assess the quality of the AI-generated research, we built an automated reviewer using o4-mini[57]. This component was designed to emulate the peer-review process of a top-tier machine learning conference by adhering to the official NeurIPS reviewer guidelines. The agent processes the PDF of a manuscript to produce a structured review, including numerical scores for soundness, presentation and contribution, along with a list of strengths and weaknesses and a preliminary accept or reject decision (Supplementary Information section A.3). All prompts used for The Automated Reviewer are provided in Supplementary Information section A.3.1.

**Review process.** The Automated Reviewer follows a multistage process. First, the system is prompted with the role: 'You are an AI researcher who is reviewing a paper that was submitted to a prestigious ML venue.' The review prompt provides the paper content along with detailed NeurIPS reviewer guidelines and asks for a structured JSON response, including a summary, strengths, weaknesses, questions,

limitations, ethical concerns and numerical scores (soundness, presentation, contribution, overall score 1–10 and confidence level). To improve robustness, the final assessment is a meta-review that ensembles five independent reviews. The five reviews are generated for each paper and aggregated into a single meta-review, with an LLM taking the role of an area chair to find consensus among the individual reviews.

**Validation.** We benchmarked The Automated Reviewer against human decisions using ICLR data from the publicly available OpenReview dataset[33]. The Automated Reviewer achieved a comparable balanced accuracy with humans (69% versus 66%; see Supplementary Information section A.3.2 for details) and a higher $F_1$ score compared with inter-human group agreement (0.62 versus 0.49) in the NeurIPS 2021 consistency experiment[34], for which roughly 10% of submissions were randomly selected and sent to two independent review committees, thus providing a real-world benchmark of inter-reviewer consistency (Table 1). These results indicate that LLM-based agents can provide valuable feedback that aligns with the opinion of the average human expert. We highlight that there was a different set of paper submissions in the ICLR and NeurIPS paper pools and, thus, a shift in the distribution, so that this comparison is not exact. However, ICLR is the only main machine learning conference that releases all accept and reject decisions, which allowed us to perform the analysis, and the NeurIPS 2021 experiment is the only modern version of the human consistency experiment, and is, thus, the only possible comparison.

## Ethics approval

This study received ethics approval from the University of British Columbia Behavioral Research Ethics Board (Protocol No. H24-02652). The research was conducted in full cooperation with the ICLR conference leadership and the relevant workshop organizers. In accordance with the approved protocol, human participants (peer reviewers) were informed that a small number of submissions to the workshop were AI-generated, although not which specific papers. Participants had the option to opt out of reviewing any potentially AI-generated manuscripts. All AI-generated submissions were withdrawn following the review process, regardless of the outcome.

## Data availability

For the nanoGPT[58] experiments, the template-based version of The AI Scientist used the Shakespeare character[59], enwiki8 (ref. 60) and text8 (ref. 61) datasets. The template-free version of The AI Scientist used the Crop Pest and Disease Detection dataset[62] for one of the papers submitted to the ICLR workshop experiment and the Waterbirds[63] and CelebA[64] datasets for the experiments shown in Figs. 1b and 3b,c. In all other cases, the template-free version used datasets available through the HuggingFace Hub[55].

## Code availability

The code for the template-based version of The AI Scientist and The Automated Reviewer is available at https://github.com/SakanaAI/AI-Scientist. Additionally, the code for the template-free version of The AI Scientist is available at https://github.com/SakanaAI/AI-Scientist-v2. Both code repositories are licensed under the Apache License 2.0.

49. Wang, L. et al. A survey on large language model based autonomous agents. *Front. Comput. Sci.* **18**, 186345 (2024).
50. Brown, T. et al. Language models are few-shot learners. In *Proc. Advances in Neural Information Processing Systems* Vol. 33 (eds Larochelle, H. et al.) 1877–1901 (Curran Associates, 2020).
51. Shinn, N., Cassano, F., Gopinath, A., Narasimhan, K. & Yao, S. Reflexion: language agents with verbal reinforcement learning. In *Proc. Advances in Neural Information Processing Systems*, Vol. 36 (eds Oh, A. et al.) 8634–8652 (Curran Associates, 2024).
52. Gauthier, P. aider. *GitHub* https://github.com/paul-gauthier/aider (2024).
53. Vaswani, A. et al. Attention is all you need. In *Advances in Neural Information Processing Systems* (eds Guyon, I. et al.) Vol. 30, 5998–6008 (Curran Associates, 2017).
54. Stanley, K. O., Lehman, J. & Soros, L. Open-endedness: the last grand challenge you've never heard of. *O'Reilly Radar* https://www.oreilly.com/radar/open-endedness-the-last-grand-challenge-youve-never-heard-of/ (2017).
55. Wolf, T. et al. Transformers: state-of-the-art natural language processing. In *Proc. 2020 Conference on Empirical Methods in Natural Language Processing: System Demonstrations* (eds Liu, Q. & Schlangen, D.) 38–45 (ACL, 2020).
56. El-Kishky, A. *OpenAI o1 System Card* https://api.semanticscholar.org/CorpusID:272648256 (OpenAI, 2024).
57. *Introducing OpenAI o3 and o4-mini* https://openai.com/index/introducing-o3-and-o4-mini/ (OpenAI, 2025).
58. Karpathy, A. NanoGPT. *GitHub* https://github.com/karpathy/nanoGPT (2022).
59. Karpathy, A. The unreasonable effectiveness of recurrent neural networks. *GitHub* https://karpathy.github.io/2015/05/21/rnn-effectiveness/ (2015).
60. Hutter, M. *The Hutter Prize* http://prize.hutter1.net (2006).
61. Mahoney, M. *About the Test Data* http://mattmahoney.net/dc/textdata.html (2011).
62. Mensah, P. K. et al. CCMT: dataset for crop pest and disease detection. *Data Brief* **49**, 109306 (2023).
63. Sagawa, S., Koh, P. W., Hashimoto, T. & Liang, P. Waterbirds. *GitHub* https://github.com/kohpangwei/group_DRO (2019).
64. Liu, Z., Luo, P., Wang, X. & Tang, X. Deep learning face attributes in the wild. In *Proc. 2015 IEEE International Conference on Computer Vision* (eds Ikeuchi, K. et al.) 3730–3738 (IEEE, 2015).

**Acknowledgements** We thank I. Zhang, J. von Oswald, T. Akiba, Y. Tang, K. Nakago, K. Misaki, H. Goda, Y. Inoue, A. Dharna, B. Norman, J. Zhang, A. Olerinyova and F. Muecke-Wegner for helpful feedback. This work was supported by grants from Schmidt Futures, NSERC, the Vector Institute, the Canada CIFAR AI Chairs programme and a donation from R. Cosman.

**Author contributions** Four authors contributed equally to this work and are listed in alphabetical order. R.T.L. (shared first) co-initiated and co-led the project and contributed core ideas. R.T.L. conceived and coded core parts of The Automated Reviewer, tailored the paper-generation pipeline to the workshop and ran the paper-generation experiments. R.T.L. organized the workshop communication process. R.T.L. read and validated the work of many AI-generated papers to select submissions and checked the implementations of the code in the papers. R.T.L. led the writing of the paper and wrote detailed analyses of the submitted papers for our article. Chris Lu (shared first) initiated and co-led the project. Chris Lu conceived the original idea and structure for The AI Scientist and developed the first working system, which demonstrated autonomous end-to-end paper generation. Chris Lu conceived the experimental set-up and performed the evaluations in the original preprint. Chris Lu led the development of the template-based version of The AI Scientist. Chris Lu ran the paper-generation experiments, led the writing of the paper, and provided advice, feedback and writing. Cong Lu (shared first) initiated and co-led the project and conceived the original ideas for The AI Scientist, including the use of software engineering agents such as Aider to execute scientific ideas autonomously. Cong Lu coded core parts of the idea generation, The Automated Reviewer, tool use, experimental aggregation and paper-writing framework. Cong Lu wrote and led the IRB approval process for the workshop experiments and evaluated AI-generated paper submissions. Cong Lu led the writing of the paper. Y.Y. (shared first, equal corresponding) co-led the project and contributed core ideas. Y.Y. coded the core tree-search and the template-free version of The AI Scientist. Y.Y. ran the paper-generation experiments. Y.Y. read and validated many AI-generated papers to select submissions and checked the code implementations in the papers. Y.Y. led the writing of the paper and wrote detailed analyses of the submitted papers for our article. S.H. enhanced the iterative Automated Reviewer with VLM feedback, contributed to the experiment and paper-writing framework, ran The Automated Reviewer benchmark and ablation experiments, helped run the paper-generation experiments, read and validated many AI-generated papers to select submissions, and checked the implementations of the code in the papers. S.H. helped write and iterate over drafts of the paper and helped write the IRB approval. J.F. provided advice, feedback and writing. D.H. (equal corresponding) provided overarching guidance for the research project and offered technical insights, advice, feedback and writing. D.H. oversaw the public communication process. J.C. (equal corresponding) provided overarching guidance for the research project and offered technical insights, advice, feedback and writing. J.C. oversaw the IRB application process and evaluated AI-generated paper submissions.

**Competing interests** The principal investigator, J.C., has affiliations with the Vector Institute and Google DeepMind. This project has Vector Institute affiliations but is not connected to Google DeepMind. Chris Lu, Cong Lu, R.T.L., Y.Y., S.H. and D.H. are employees of or consultants for Sakana AI, a machine learning research company involved in the design of The AI Scientist. The principal investigator is not financially compensated by Sakana AI. These arrangements have been reviewed and approved by the University of British Columbia in accordance with its conflict of interest policies. The other authors declare no competing interests.

**Additional information**
**Correspondence and requests for materials** should be addressed to Yutaro Yamada, David Ha or Jeff Clune.
