## [Peer Review File · Nature]

nature portfolio

Peer Review File

Towards End-to-end Automation of AI Science with AI

Corresponding Author: Professor Jeff Clune

Version 1:

Reviewer comments:

Referee #1

(Remarks to the Author)

This is good and interesting work with some serious framing issues.

To begin with, the title is unreasonably overselling the paper. It reads more like clickbait than like the title of a scientific paper. The second sentence of the abstract echoes this, by claiming that AI can automate the whole scientific process. The idea that these systems could "radically accelerate the pace of scientific discovery, ushering forward an era that could see the solution to many of society's greatest challenges" also seems premature at the very least. It is a very long way from producing a convincing simulacrum of a scientific paper to actually accelerating the pace of scientific discovery; in fact, one might argue that the generation of even more mediocre papers could slow science down, rather than the opposite. Saying that the system described in this paper represents "full end-to-end automation of science" seems about as accurate as saying that Midjourney represents the end-to-end automation of art. (For the avoidance of any doubt, this means: not very accurate at all; in fact entirely inaccurate.)

What the paper actually shows is that there exists a system which can produce a mediocre workshop paper with limited human intervention (human screening of generated ideas). This is very impressive in itself, and this impressive result deserves to be discussed on its own merits. The fact that an LLM-based agentic system can now produce a basic scientific paper, for better or worse, is important to highlight and should spark a necessary discussion about the scientific ecosystem and the nature of science. In this context, the current bombastic language rather serves to detract from the actual contribution. I think Nature should eventually publish this paper, but only after these claims have been toned down considerably.

Another way of putting this is that I, the reviewer, am convinced that the work is good and important and should be published. Well done, authors, you have succeeded. Now please stop sounding like a used car salesman or second-rate politician, and rewrite the appropriate parts of the paper in the manner of scientists talking to other scientists. Be open about what the system can and cannot do, which capabilities are likely to develop fast with better models, and which capabilities are likely to be harder to improve. Also, no need to hide (in the supplementary material) that humans curated the research ideas or that many of the ideas were quite similar to each other.

Technically, the paper is quite solid. I can't find any major shortcomings, and the supplemental material is as far as I can see complete.

I think it should be clarified in the main paper text that the generated paper that passed peer review had a negative result. There is nothing wrong with that - negative results are important - but I think it should be pointed out.

The section C.3, limitations and broader impact, is interesting for what it addresses as well as for what it doesn't address. It is welcome that the issues with overreliance on the review system with automatically generated papers are pointed out. This could be explored in more depth, as I think the issue of "clogging the pipes" of scientific peer review is pressing already as it is. But there is a further problem. Imagine that some future AI scientist actually becomes as good, or better, as a good human researcher at ideating and executing experiments. What, then, becomes of human scientists? The authors of the paper under consideration are AI researchers; presumably they do not want to become replaced by their own creations. Or do they? What effects would this have on trust in science and in human expertise, incentives for learning, and the social fabric? I think this is an important enough issue to be discussed further in the supplemental material and also at least mentioned in the main paper text.

(Remarks on code availability)

Response to Referee #6 (Remarks to the Author):

I co-reviewed this manuscript with one of the reviewers who provided the listed reports.

Referee #6 (Remarks on code availability):

The repository is a well-documented codebase that successfully implements and demonstrates the core methodology described in the paper. It is an excellent resource for the community to understand, experiment with, and build upon the "AI Scientist" paradigm.

We appreciate the time you spent looking at the code. Thank you for the kind words. We are delighted you agree this is a valuable resource for the community.

Referee #2

(Remarks to the Author)

Summary

This paper describes a novel AI agent system for tackling the scientific research discovery process: find research question, develop research idea, run experiment to validate the idea and iterate; the final outcome of the process is to present a research paper.

To mimic the crucial peer review process in science, the authors designed an automated review system to provide feedback following machine learning conference guidelines.

The overall framework includes two settings: one called template-based AI scientist where human scientists initiated the system by providing some codes and experiment setups and the other called template-free AI scientist where the discovery is done without the initialization.

One of the generated papers was accepted to a workshop in the ICLR conference (one of the major machine learning conferences) through the blind review process.

Comments

First of all, this is one of the first work that considers fully automated AI system in an "end-to-end" manner: from idea generation to peer review and finally present research outcome in a research paper, and leads to peer-reviewed publications. I reviewed the method section, and the proposed methodology is technically sound. My main concerns are on the experiment side and if the results support the claim and future promise.

Questions

Q1: This paper mainly presents one paper generated by the agent system accepted for an ICLR workshop related to challenges and limitations in applied deep learning. I noticed the scope narrowed down suggested by the editors, but I'm still concerned if one case study would represent machine learning research entirely---as noted by the authors, the workshop received only around 40 submissions. The paper could benefit from more study on different category of machine learning work.

Q2: Even if a paper passes conference peer review, the claim about automating discovery could be made more cautious. For example, even from human scientists, there are also a significant amount of incremental work to improve some certain technical parts or putting two architectures together to create a new one. These types of borderline papers can be accepted by chance. It would be good to see a comprehensive analysis of the types of research AI agent would generate.

Q3: The authors justify the future promise of the framework by demonstrating stronger models with more computational resources could improve the performance measured by the automated reviewer. As noted by the authors, workshops often have around 70% acceptance rate (way higher than 25-30% acceptance rate in main conference). In addition, the paper is significantly shorter: 4 vs 8-9 pages. The reviewers often evaluate workshop papers in a much lighter way compared to conference submissions. How do the authors justify the gap will be bridged by the improved compute or model performance? Stronger evidence would be extremely helpful for this claim to be made to such broad audience in Nature.

Q4: Can you clarify how exactly human is involved in the template-based setting? Does human also formulate the problem? E.g. in your first example, before the agent system proposed the idea, do you start by giving the code for some diffusion models and prompting it to improve them? I assume yes as it is unlikely to provide code/experiment without formulating the problem.

Q5: For the generated papers accepted in the workshop, it is from the template-free approach. Did you also submit the papers generated by the template-based approach?

Q6: For the evaluation of the automated review system, the NeurIPS experiments spotted the issues of human reviewers as two batch of reviewers could lead to opposite decisions. Showing the overall accuracy in agreeing with the decision of one group of reviewers is great, but do you spot some patterns that can categorize LLM generated reviews?

Overall, this work is the first to consider fully automated agent for scientific research discovery, but I'm mostly concerned about the limited quantitative analysis and the strong enough evidence to support the claim of "entirely automating" as a message this paper would send to the community.

Relevant work

Two related works (later than AI Scientist-V1, but concurrent to AI Scientist-V2) also claimed generating papers accepted in ICLR 2025 workshops: Carl <https://www.autoscience.ai/blog/ai-research-automation> and Zochi <https://www.miloby.ai/blog/zochi-tech-report>. Zochi also later announced another paper first-time passed the peer review of ACL main conference

(Remarks on code availability)

Referee #3

(Remarks to the Author)

The papers reports on an implementation of a automated process to perform ML research using LLM agents. The process involves formulating a hypothesis, generating, debugging code to perform computer experiments, finetuning hyperparameters, generating plots, assessing visualizations, performing ablation studies and writing a paper in Latex which is auto-reviewed (and meta reviewed) by an ensemble of LLMs.

captured as structured notes. We have added this description in Supplementary Section A.2.4.

7. Lines 400–401: The authors generate related work only after the experiments are completed and during manuscript preparation. It would make more sense to integrate this step into the literature review stage.

We separate the novelty check and literature review as distinct tasks. The former ensures that the idea is novel before running experiments, while the latter contextualizes results afterward. Although this means that the AI Scientist searches the literature twice, their purposes differ. However we agree that this could be done differently and better. That is an interesting area for future work.

8. It would be helpful if the paper included a detailed comparison of this work with existing Deep Research systems.

Thank you for this helpful suggestion. While systems like Deep Research are powerful tools for synthesizing and collating pre-existing literature to provide well-researched answers to a user's query, it cannot conduct any experiments or investigate new ideas. The AI Scientist automates the entire scientific process, from originating a novel research idea through experimentation and finally writing a complete scientific manuscript. We added this information to the Supplementary Section A.2.5.

9. Lines 568–580: How was the balanced human accuracy calculated? Was a group of human reviewers invited to provide the baseline evaluation?

We compute balanced accuracy as the average of per-class recall for accept and reject. We did not recruit human reviewers for this calculation. The statement "The Automated Reviewer achieves a comparable balanced accuracy with humans (69% vs. 66%)" refers to the data from the NeurIPS 2021 reviewer-consistency experiment. From the aggregated results of two human reviewer pools, we treat one pool as the reference and evaluate the other pool against it (Accept = {Oral, Spotlight, Poster}; Reject = {Reject}).

This yields the human-human confusion matrix and the following induced 2x2 counts:

True-Accept predicted Accept = 99,

True-Accept predicted Reject = 107,

True-Reject predicted Accept = 96,

True-Reject predicted Reject = 462

Therefore, Sensitivity (Accept) = $99/206 = \sim 0.481$, Specificity (Reject) = $462/558 = \sim 0.828$, and Balanced human accuracy = $(0.481 + 0.828)/2 = \sim 0.655$.

This description is added in Supplementary Section A.3.2

3 papers were produced and submitted to a ICLR conference workshop. It was found that one of them was above the acceptance bar justifying the claim that this could be considered the first fully automated publishable scientific paper.

The main contribution seems to be the design and implementation of a workflow of various kinds of LLMs from OpenAI and Anthropic, that can autonomously conduct research. These LLMs need to coordinate with each other and with internet resources and datasets on HuggingFace, providing them with the correct context information in their prompt to perform their research.

I find the paper not very original in the sense that this is a rather obvious question to try to answer. The approach to achieving this result is however somewhat novel.

I found it interesting to see what the state of the art is in fully automated science, and in this sense I think this is an interesting status report. I was less impressed with the actual result: find the bar of passing a workshop review process rather low. It would have been a lot stronger if their would have been a genuinely interesting discovery or innovation, but this is not reported.

In fact, I read the actual paper and it did not make a whole lot of sense to me. The sentences read like an ML paper but is it more than a correct sounding paper? I can see how this gets accepted into a workshop with the title "I can't believe it's not better", because it didn't work really well. But besides a low bar for papers that report on research that didn't end up working very well, I actually don't think what was proposed made much sense. It would have been nice if the authors would have given their own assessment of the written paper.

This brings me to my final point. Are we ready to unleash AI scientists into the world? The authors are well aware of the dangers of flooding the internet with bad science and they do reflect on it. However, the availability of these tools will undoubtedly have the effect that many young researchers whose careers depend on papers in top tier conferences will give this a spin. In that sense the paper does us a service to let us know where we stand and what to expect.

The methodology and use of statistics seems sound. The paper could be clearer in the sense that a more concrete example of what the process delivered in every phase would be welcome. Also I think there is too much emphasis in the main paper in the artificial reviewer which I think is not the main achievement. More details on the workflow would have been better.

In summary, I would say that, while interesting, the paper does not convincingly show that automated AI research is at a level where it can produce interesting research. The paper that was supposedly accepted was very far from an interesting discovery. And the contribution of this submission, namely the design of an agentic workflow to conduct automated research, was interesting but not quite enough to merit publication in Nature. I also think the authors could have done a better job at critically assessing the actual scientific contribution made but the system.

(Remarks on code availability)

Releree #4

(Remarks to the Author)
Summary:

- The authors propose an AI system that autonomously performs all steps in scientific research from conception all the way to publication.
- The first phase is idea generation: the agent creates an archive of high-level research ideas and selects promising directions. It then filters ideas by checking Semantic Scholar to ensure they are not too similar to existing publications.
- The next phase is experiment execution, using two approaches: template-based (starting from existing code) and template-free (writing a starter script and iteratively improving it). After each experiment, the agent records notes in an experimental journal.
- The third phase produces a formal LaTeX write-up, which is reviewed by an Automated Reviewer that outputs numerical scores and an accept/reject decision. The reviewer was evaluated using past submissions to a machine-learning conference.
- Authors submit 3 papers generated by their agent (with guidance provided by authors) to a workshop at ICLR. 1 out of these 3 passed the threshold for acceptance.

Comments:

Results seem premature:

- The authors are exploring an important direction; their early approaches show promise and are worth pursuing further. However, at this stage, the submitted paper, particularly in making the strong claim of fully automating science, is not suitable for publication in Nature.
- Line 66: "marking the first instance of a fully AI-generated paper successfully navigating a peer review." In the ML community, workshops are primarily venues for disseminating and discussing early work rather than rigorously peer-

manuscript rather than every intermediate step. This alignment helps maintain comparability between human and automated review processes.

We also agree that grounding evaluations in literature would benefit the reviews. In our current setup, we rely on the background knowledge already embedded in the LLM through training on web-scale data which typically includes scientific contributions up to the LLM's cutoff date. Explicitly integrating literature searches into the review process is a promising direction for future versions of the system. We have mentioned both of these points in Supplementary Section A.4.2.

4. At different stages of the pipeline, the capability requirements for the models vary. The paper mentions that different large models are used in different parts of the workflow. How are the appropriate models selected for each stage?

In our current implementation, we empirically selected the best model for each role through small-scale trials (e.g., Claude performing best for code generation, GPT models for planning and analysis). These closely align with publicly available benchmark scores for these types of capabilities (e.g. LMSYS Leaderboard), meaning one could pick models without any experimentation to save time and compute.

Over time, as foundation models continue to improve, we expect these stage-specific distinctions to diminish and a single sufficiently capable model or agentic system could handle the entire workflow. However, if it remains the case that certain models are better at different stages, our system is designed to flexibly accommodate such substitutions without structural changes. We have added this information in Supplementary Section A.2.3.

5. Lines 556–566: Are the independent reviews produced by using different models or if we different prompts, or are they generated by the same model with identical prompts through multiple sampling runs?

Independent reviews are produced using the same model with identical prompts through multiple sampling runs with different stochastic sampling, which leads to a diversity of different reviews and scores. We aggregate these independent reviews into a final meta review using the same model, and show in Table A4 that empirically this leads to the closest calibration to human reviewers. We have added this description in Supplementary Section A.4.2.

6. Line 392: How is the experimental journal organized? Is it structured in a specific format?

In the template-based setting, the experimental journal was a simple text file. In the template-free setting, the experimental journal is implemented as a Python class that produces a structured JSON export. It records every node in the tree, with each node containing generated code, a plan, experimental results, and LLM/VLM feedback etc.

reviewed outlets. Thus, it is not reasonable to claim that the paper underwent the full scientific peer-review process.
- A core challenge at ML conferences is randomness in reviewer assignment. Given high submission volumes, there are often too few qualified reviewers, and this problem is exacerbated for satellite workshops. While acceptance at a major ML conference could help demonstrate viability, the authors should show performance across multiple submissions. In this context, only 1 of 3 workshop acceptances does not provide enough signal to distinguish random reviewer noise from genuine innovation.

- The paper itself is not impressive: it is a 4-page submission describing a simple experiment—adding an additional regularization term to the loss function—that ultimately was not useful. The experiment may be informative, but it does not suggest a scientific breakthrough, even if largely developed by an AI agent.

- In a real conference or journal submission, there would be at least one additional round of peer review which involves responding to and rebutting the arguments proposed by the reviewers. This process was not simulated here.

- Lastly, the authors themselves seem to indicate that this may be a premature submission as they state in Line 173: "The AI scientist does not yet meet the standards for top tier publications, nor even consistently for workshops" and in Line 165 "In assessing the impact of a technology, it is thus important to keep in mind its likely future trajectory." Indeed there is strong potential in the field of automated research and scientific agents, but this paper does not show clear evidence that we have reached that stage yet.

- It should also be made clear whether other submissions were also made by this agent to venues previously. The code repository linked by the paper appears to show many additional generated papers (>50?), were those also submitted? How many attempts have been made? With enough random tries, acceptance becomes unsurprising. If so, what were the outcomes of those experiments?

- The agent is not really fully autonomous: As described by the authors, there are several manual filtering steps involved. Lines 273-286 highlight how human curation and filtering was required at multiple stages from idea filtering to scrutinizing experiments and removing failed runs, as well as reviewing final papers for integrity of citations. I think it's not accurate to claim that these papers were entirely generated by AI.

- Deeper analysis on the performance of automated reviewer would be useful. In Line 113 the authors claim that "This demonstrates [the automated reviewer's] ability to replicate the collective judgment of human reviewers with high fidelity". However, it's unclear if this is true given that only 1 out of 3 of their submissions to a machine learning workshop venue were scored above the acceptance threshold. It's surprising that such papers made it past their automated reviewer which was evaluated on paper from the NeurIPS conference (Table 1).

- The most novel capability presented by the agent appears to be generation of novel hypotheses. Further analyses and details on the agent's performance on this task and how it was improved would be useful. Once given a core prompt or scientific question, there are already several examples of AI tools doing well in executing and experimenting with different ideas.

(Remarks on code availability)

Relesee #5

(Remarks to the Author)

This work proposes an agent pipeline designed to fully automate the research process in the field of ML/AI science. Notably, the authors demonstrate that one of the manuscripts produced by "The AI Scientist" system successfully passed peer review to reach the acceptance level at a major AI conference. This suggests that the approach can achieve human-level performance in fully automated research.

In general, I find the idea of automating the entire research workflow highly promising, from literature review and idea proposal to experiment execution and manuscript writing. Such an approach holds great potential to accelerate the scientific discoveries and transform the way research is conducted in the future. However I still have a few remarks and concerns in the current work:

1. The AI Scientist workflow involves multi-stage generation. Are the execution histories of earlier stages accessible to later stages, and are they explicitly used to guide subsequent generation? If so, what is the detailed mechanism? For instance, is there a unified memory system that connects different stages? If not, how does the system ensure consistency across the entire pipeline, from idea initialization to the final manuscript, given that current LLMs are prone to "hallucinations"? Otherwise, the final paper may suffer from error propagation, diverging from the initial proposal or experimental results.

2. The system architecture of the pipeline is not clearly described. Is the execution between stages fully automated, or does it require human intervention to transition between stages (e.g., executing experimental code)? If it is automated, how is the system architecture designed? Particularly the tool call APIs, automated execution, sandboxed execution, and its interactions with the operating system or model training components.

3. Regarding the design of the "Automated Reviewer", the authors note that it only takes the manuscript PDF as input, thereby focusing solely on the quality of the final paper. I think it would be more informative to also evaluate the intermediate outputs of the pipeline. In addition, a professional and responsible reviewer is expected to be familiar with the relevant background literature in order to properly position the paper within its research field. I think such Reviewer module should therefore incorporate related literature to contextualize its assessment; otherwise, the evaluation may lack sufficient

Particularly the tool call APIs, automated execution, sandboxed environments, and its interactions with the operating system or model training components.

Transitions between different stages are fully automated (no human intervention is required). A single driver process orchestrates the experiment pipeline: it generates candidate programs via LLM calls, launches experiments, monitors exit codes/logs/timereouts, and triggers state transitions (between Stages 1-4, which go through Preliminary Investigation -> Hyperparameter Tuning -> Research Agenda Execution -> Ablation Studies) after evaluating the pre-defined transition criteria:

- Stage 1 -> Stage 2: the system must produce runnable code with no runtime errors.
- Stage 2 -> Stage 3: the code must outperform the root/baseline under the metrics (which are defined by LLM before Stage 1 but after ideation.)
- Stage 3 -> Stage 4: triggered automatically once the Stage 3 search exhausts its allocated nodes. (These criteria are detailed in the Methods section.)

Each experiment is launched programmatically via Python's subprocess (non-interactive) module for running external commands, and the driver process monitors exit codes, logs, and timeouts. After each run, artifacts such as metrics, figures, any runtime errors, and VLM feedback on figures, are captured and stored in a typed Python object representing a tree node. The driver process calls an LLM judge to select the best node from a stage and passes it forward.

Tool invocations (code generation, analysis, VLM feedback) are made through internal Python-callable interfaces (which wraps LLM API calls) from the driver process; experimental code interacts with the OS only through the controlled subprocess boundary and file I/O in per-run work directories.

Thank you for highlighting that these details were not clear in the paper. To fix that, we have added these details to Supplementary Section A.2.1. These details are also all contained within our fully open-source code.

3. Regarding the design of the "Automated Reviewer", the authors note that it only takes the manuscript PDF as input, thereby focusing solely on the quality of the final paper. I think it would be more informative to also evaluate the intermediate outputs of the pipeline. In addition, a professional and responsible reviewer is expected to be familiar with the relevant background literature in order to properly position the paper within its research field. I think such Reviewer module should therefore incorporate related literature to contextualize its assessment; otherwise, the evaluation may lack sufficient grounding.

We agree that incorporating intermediate outputs could enrich the Automated Reviewer, and thus represents an interesting opportunity for new research. Our current design intentionally mirrors how human reviewers typically operate, which is evaluating the final

4. At different stages of the pipeline, the capability requirements for the models vary. The paper mentions that different large models are used in different parts of the workflow. How are the appropriate models selected for each stage?
5. Lines 556–566: Are the independent reviews produced by using different models or different prompts, or are they generated by the same model with identical prompts through multiple sampling runs?
6. Line 392: How is the experimental journal organized? Is it structured in a specific format?
7. Lines 400–401: The authors generate related work only after the experiments are completed and during manuscript preparation. It would make more sense to integrate this step into the literature review stage.
8. It would be helpful if the paper included a detailed comparison of this work with existing Deep Research systems.
9. Lines 568–580: How was the balanced human accuracy calculated? Was a group of human reviewers invited to provide the baseline evaluation?

(Remarks on code availability)

Referee #6

(Remarks to the Author)

I co-reviewed this manuscript with one of the reviewers who provided the listed reports.

(Remarks on code availability)

The repository is a well-documented codebase that successfully implements and demonstrates the core methodology described in the paper. It is an excellent resource for the community to understand, experiment with, and build upon the "AI Scientist" paradigm.

Version 2:

Reviewer comments:

Referee #1

(Remarks to the Author)

I have read the new version of the manuscript, and it is much improved, in particular in terms of tone and realistic claims. I was already convinced of its technical quality, so I'm happy to recommend that the manuscript is published as is. Congratulations on writing a good and important paper!

This being said, I don't like... the results. I don't like that current AI systems are this capable of doing science, and upcoming systems will be even more capable. As a researcher, I would like us humans to remain in the driving seat, and research to be done by humans, using whatever tools we have at our disposal. Results such as yours make a future thinkable where humans are no longer in the driving seat, or even relevant, for the scientific process. I dearly hope it will never come to that. However, the results are what the results are, and it is good that the world knows this.

Julian Togelius

(Remarks on code availability)

Referee #2

(Remarks to the Author)

First of all, I appreciate the authors toned down the main claims of the paper. It is helpful to shape the message sent to the community.

Adding the discussions about the two relevant works is helpful.

I respect the optimism the authors demonstrate towards improved LLMs and thus improved AI scientist workflow which I agree, could be true. However, I have to say, there is no strong evidence that the scaling still holds for base LLMs as well as it would hold in what way for AI scientists (e.g. discussions would be helpful here, fine-tuning, reasoning, inference-time scaling). I am not arguing that it is not the case, but I would like to avoid over-optimism given the generated papers are only okay and only passing the bar of workshops. As a machine learning researcher, I can say a lot about the process, but I know the bar with randomness and kindness for workshop papers. I am not arguing there was no progress, but I personally would not draw too strong conclusions based on current results.

I have another question regarding the possibility of hallucination by LLMs, and think it is helpful to add some discussions about it in the paper/supplementary material. Since we know LLMs sometimes hallucinate severely, do the authors spot any relevant issues in the study?

Minor: The std is not denoted in the table 1's caption, figure 2 (B), (C) and figure 3 (C) does not explain the error bars.

This work proposes an agent pipeline designed to fully automate the research process in the field of ML/AI science. Notably, the authors demonstrate that one of the manuscripts produced by "The AI Scientist" system successfully passed peer review to reach the acceptance level at a major AI conference. This suggests that the approach can achieve human-level performance in fully automated research.

In general, I find the idea of automating the entire research workflow highly promising, from literature review and idea proposal to experiment execution and manuscript writing. Such an approach holds great potential to accelerate the scientific discoveries and transform the way research is conducted in the future.

We appreciate the time you are spending reviewing our paper. We are delighted to hear you agree that systems like The AI Scientist "hold great potential to accelerate the scientific discoveries and transform the way research is conducted in the future."

However I still have a few remarks and concerns in the current work:

1. The AI Scientist workflow involves multi-stage generation. Are the execution histories of earlier stages accessible to later stages, and are they explicitly used to guide subsequent generation? If so, what is the detailed mechanism? For instance, is there a unified memory system that connects different stages? If not, how does the system ensure consistency across the entire pipeline, from idea initialization to the final manuscript, given that current LLMs are prone to "hallucinations"? Otherwise, the final paper may suffer from error propagation, diverging from the initial proposal or experimental results.

Each experiment stage exports its best node to the following stage, which then uses it to guide subsequent generation. The transition from ideation to experimentation is handled by passing the finalized idea JSON, which contains a title, hypothesis, related work, experiment plans, risk factors and limitations (See Supplementary Section A.2.7 for an example idea JSON file). The transition from experimentation to writing is handled by passing a condensed version of the experiment journal, which contains the best node from Stage 3 (research agenda execution) and all nodes in Stage 4 (ablation studies). This mechanism ensures that later stages remain grounded in the outputs of earlier stages and helps reduce divergence between the initial proposal, results, and final manuscript. We have clarified these details in the updated Supplementary Section A.2.2.

We acknowledge that there is still an issue of hallucination especially during the writeup stage e.g., a paper referencing results present only in the experimental log and not in the paper, but we expect such issues to diminish as LLM reasoning and factual consistency continue to improve.

2. The system architecture of the pipeline is not clearly described. Is the execution between stages fully automated, or does it require human intervention to transition between stages (e.g., executing experimental code)? If it is automated, how is the system architecture designed?

(Remarks on code availability)

Referee #3

(Remarks to the Author)

I have read the updated paper and the rebuttal to my review. I appreciate the fact that some of the claims were toned down, as many reviewers have requested this.

I think the paper represents an interesting progress report on where we stand when we put current SOTA LLMs to work to do science. There is a clear technical contribution in the "plumbing together" of these LLMs in a sensible way to go through an entire scientific research project, with agentic reviewers and authors and all.

I have however a hard time seeing a breakthrough scientific contribution worthy of a Nature publication, but I may be miscalibrated on what Nature likes to publish. The paper will definitely find many interested readers and spur discussion in the field, so maybe for that reason alone it is worthy to publish.

(Remarks on code availability)

Referee #5

(Remarks to the Author)

Thank you for your detailed and thoughtful responses to my comments. I appreciate the clarifications provided, particularly regarding the system architecture, inter-stage coordination, and the model selection process. The revisions made to the manuscript and the supplementary materials have adequately addressed my technical concerns. The added details significantly improve the clarity and rigor of the work. While the acknowledgment of lingering challenges like hallucinations during the write-up stage is noted, I believe the authors have been sufficiently transparent about the current limitations of the system.

(Remarks on code availability)

Referee #6

(Remarks to the Author)

I co-reviewed this manuscript with one of the reviewers who provided the listed reports.

(Remarks on code availability)

I've noticed that the repository has not been updated for a while and there are several open issues without responses. To maximize the long-term value and impact of your great work, it would be incredibly beneficial if the authors could engage with these open issues and provide occasional maintenance.

The apparent discrepancy arises from the target venue context: when using the AI Reviewer with its default prompt tuned for top-tier conference acceptance, all three submissions were indeed rated below the bar, which is consistent with the workshop reviewer's observation. However, when we changed the prompt to reflect workshop-level review standards, one of the three submissions (the same paper that was accepted at the ICLR workshop) was judged to meet the acceptance bar. We have updated the manuscript to include this important detail. Specifically, we added the following text in Supplementary Section A.4.2:

"These statistics are for aggregate human consensus on the NeurIPS-2021 consistency dataset [7], a benchmark reflecting top-tier conference standards. Porting a conference-calibrated reviewer to a *workshop* setting requires prompt (and threshold) recalibration. In our ICLR workshop experiments, the Automated Reviewer with its default "top-tier conference" prompt rated all three submissions below the acceptance bar for conferences. After explicitly calibrating the prompt for a workshop setting, one of the three—the paper ultimately accepted at the ICLR workshop—was judged above the bar."

- *The most novel capability presented by the agent appears to be generation of novel hypotheses. Further analyses and details on the agent's performance on this task and how it was improved would be useful. Once given a core prompt or scientific question, there are already several examples of AI tools doing well in executing and experimenting with different ideas.*

The paper contains our extensive qualitative analysis of 4 generated papers, including commentary on the level of creativity of the ideas. Also, we have added the case mentioned above where the system produced an idea that was independently pursued by another team, which resulted in a paper published in a journal. Of course, there is also the evidence that one paper was accepted at a workshop. While there may be systems that either require substantial manual intervention or execute and experiment with different ideas in small pieces of the overall pipeline of producing a paper, ours was the first (and in our view still the best by a large margin – we have since added a discussion on this in Appendix Section A.4 thanks to the reviewer's comment) at autonomously performing the entire process from idea generation through writing a paper (and evaluating it via review), which is an important component of the advance this work describes.

Thank you again for your review. We have endeavored to address all of your concerns, and the paper is stronger as a result. We hope you agree it is now ready for publication.

Response to Referee #5 (Remarks to the Author):

In cases where reviewers are anonymous, credit should be given to 'Anonymous Referee' and the source. The images or other third party material in this Peer Review File are included in the article's Creative Commons license, unless indicated otherwise in a credit line to the material. If material is not included in the article's Creative Commons license and your intended use is not permitted by statutory regulation or exceeds the permitted use, you will need to obtain permission directly from the copyright holder. To view a copy of this license, visit <https://creativecommons.org/licenses/by/4.0/>.

think this is the "GPT-1" moment for AI scientists, and the GPT-1 paper was a major, important milestone to share with the community.

- It should also be made clear whether other submissions were also made by this agent to venues previously. The code repository linked by the paper appears to show many additional generated papers (>50?), were those also submitted? How many attempts have been made? With enough random tries, acceptance becomes unsurprising. If so, what were the outcomes of those experiments?

Only three papers were submitted for review, so it is not the case that others were reviewed and rejected (i.e. there are no unreported rejections). We did not have the papers go through peer review at other venues. Thank you for pointing out that we should have made that clearer. We have added the following sentence to the main text: "This was the only venue that we submitted to."

The 50 papers you mentioned are to give interested readers and the scientific community a large set of papers produced by The AI Scientist to evaluate and analyze.

- The agent is not really fully autonomous: As described by the authors, there are several manual filtering steps involved. Lines 273:286 highlight how human curation and filtering was required at multiple stages from idea filtering to scrutinizing experiments and removing failed runs, as well as reviewing final papers for integrity of citations. I think it's not accurate to claim that these papers were entirely generated by AI.

We have toned down the rhetoric substantially, which we hope addresses your concerns. Even without manual filtering, these papers would have been generated exactly in their final form. It would just be that many other papers also would exist (which could have been submitted, but we did not want to overwhelm the review system). Thus, the papers were entirely generated by AI, alongside other papers we did not choose. However, to err on the side of being conservative and at the request of you and other reviewers, we have significantly reduced the tone of these claims. In the main text, we now write: "The overall process was then run to generate ideas, experiments, and papers. We manually filtered the most promising outputs at each stage (Supplementary Section A.4). Had this filtering not occurred, the papers under analysis would still have been produced in their final form, just along with other papers and thus at a greater total cost."

- Deeper analysis on the performance of automated reviewer would be useful. In Line 113 the authors claim that "This demonstrates [the automated reviewer's] ability to replicate the collective judgment of human reviewers with high fidelity". However, it's unclear if this is true given that only 1 out of 3 of their submissions to a machine learning workshop venue were scored above the acceptance threshold. It's surprising that such papers made it past their automated reviewer which was evaluated on paper from the NeurIPS conference (Table 1).

Response to Referees and Editors

Manuscript Title: *Entirely Automating AI Science with AI*
Manuscript ID: # 2025-07-17246

Dear Yann Sweeney,

We thank the Editor for their constructive feedback and for the opportunity to revise our manuscript. Below, we provide a point-by-point response to each editorial comment. All line numbers refer to the revised manuscript unless otherwise noted.

1. Editor's comment: *Taking care to be clear about the limitations of the current system (AI research only). Extrapolation to other domains etc. to be avoided.*

Response: We have clarified the scope and limitations of the current system in two locations in the main text:

- **Lines 45–46 (main text):** "We focus on machine learning science, where experiments occur entirely on the computer."
- **Lines 186–189 (main text):** "Currently, The AI Scientist conducts computational experiments only. In future work, this same playbook could be applied to other scientific domains where one can automatically conduct experiments (or have humans conduct them) and get data back from them (e.g. automated chemistry labs, on which swift progress is being made)."

These revisions ensure that our claims are appropriately scoped to AI/ML research and that extrapolation to other domains is clearly framed as future work.

2. Editor's comment: *It would be helpful to readers to discuss the 'failure modes' of the system a bit more in the main text and figures. While some details are in the Supplementary Information, if you are showing the success case, the 'failure' case(s) should be shown too.*

Response: We have expanded our discussion of failure modes in the main text and added a new section in the Supplementary Information:

- **New Supplementary Section A.4** titled "*Human Workshop Evaluation: Paper Generation, Paper Selection, & Failure Modes*" provides detailed categorisation and examples of common failures.
- **Lines 174–185 (main text):** "Common failure modes include the generation of naive or underdeveloped ideas, incorrect implementations of the main idea, a lack of deep methodological rigor, errors in experimental implementation, duplicating figures in the

overlap underscores that AI systems are beginning to originate scientifically meaningful ideas—an early signal of what's to come.

We have added this discussion in the "Novelty and idea overlap" section in the Supplementary Section C.3.

- *In a real conference or journal submission, there would be at least one additional round of peer review which involves responding to and rebutting the arguments proposed by the reviewers. This process was not simulated here.*

That is true, but we do not claim our system can handle that piece of the scientific pipeline. One could extend our pipeline to include that phase. However, doing so requires a level of deception we were not comfortable with. We did not want to go back and forth with a reviewer without them knowing they were talking with AI. Also, importantly, the target venue for our submission was a workshop, where a formal rebuttal stage is typically not part of the process, so our evaluation faithfully mirrors the actual review procedure used in that setting and for this workshop.

We've added the following text to Appendix C.3 to clarify this point:

"Additionally, unlike human reviewers and researchers, the Automated Reviewer and The AI Scientist are currently unable to automatically perform the back-and-forth exchanges of a rebuttal or revision phase. While the pipeline could be extended to include rebuttals and/or revisions, which is an interesting area of future work, doing so would have entailed undisclosed back-and-forth with human reviewers. Moreover, because our target venue was a workshop where a formal rebuttal stage is absent, our evaluation mirrors the actual review procedure used in that setting."

- *Lastly, the authors themselves seem to indicate that this may be a premature submission as they state in Line 173: "The AI scientist does not yet meet the standards for top tier publications, nor even consistently for workshops" and in Line 185 "In assessing the impact of a technology, it is thus important to keep in mind its likely future trajectory." Indeed there is strong potential in the field of automated research and scientific agents, but this paper does not show clear evidence that we have reached that stage yet.*

We respectfully disagree with the assumption that AI-generated papers must already "meet the standards of top-tier publications" in order for a paper about an AI scientist system that produces papers to be accepted. We position this work as documenting an early capability milestone. The cited statement was meant as a transparent self-assessment, not an admission of premature submission. Our contribution lies in demonstrating the feasibility and early promise of an automated AI-driven scientific workflow, which we believe is both timely and valuable even in its formative stage. It also is important to highlight technologies about to be transformative. Indeed, it is often better to alert the community as to what transformative technologies are coming, rather than to wait until they are mature before publishing the first major paper on them. We

workshop-level standards, let alone the rigor required for top-tier conference publications.” (Supplementary Section C.3)

We have also made the tone and claims more modest. We hope all of this collectively addresses your concern.

- A core challenge at ML conferences is randomness in reviewer assignment. Given high submission volumes, there are often too few qualified reviewers, and this problem is exacerbated for satellite workshops. While acceptance at a major ML conference could help demonstrate viability, the authors should show performance across multiple submissions. In this context, only 1 of 3 workshop acceptances does not provide enough signal to distinguish random reviewer noise from genuine innovation.

We acknowledge reviewer-assignment noise, but the very fact that an AI-generated paper is within the quality range where noise matters is itself noteworthy. Moreover, as we noted above, our submission to Nature is not based solely on the fact that one paper of three was accepted. Instead, we present multiple different sources of evidence that all suggest The AI Scientist is capable of producing some papers above the threshold for acceptance at a workshop. In addition to the paper being accepted, our own extensive qualitative analyses (which we describe and include in the paper) of 4 papers confirm that it is capable of producing workshop-level work. Additionally, we release 50 AI-generated papers in total for readers to evaluate in the open-source repository (see Code Availability), and six in the appendix. We also provide quantitative evidence from the automated reviewer, which itself was calibrated and was shown to provide scores as reliably as human reviewers. The scores from the automated review are in line with the view that some of The AI Scientist papers (but not all) are above the acceptance threshold for workshops. Also, we have substantially toned down the overall rhetoric regarding the size of the advance. We believe the evidence definitely supports these more modest claims, and hope you agree.

- The paper itself is not impressive: it is a 4-page submission describing a simple experiment—adding an additional regularization term to the loss function—that ultimately was not useful. The experiment may be informative, but it does not suggest a scientific breakthrough, even if largely developed by an AI agent.

We agree that the scientific content of the generated paper itself (e.g. a regularization experiment) is modest. However, the breakthrough we emphasize is not the specific finding of the generated paper, but the fact that the system carried out the entire scientific process end to end. Moreover, we already observe early signs that the system's ideas are approaching those of human researchers: for example, one generated proposal independently came up with an idea for a paper that a later human-authored study published in *Physica D: Nonlinear Phenomena*, which investigated grokking through compression via Minimum Description Length (MDL). While the human version executed the idea more effectively (e.g., through a more creative MDL estimator), the conceptual

main text and the appendix, and hallucinations such as inaccurate citations (a full analysis of failure modes is provided in Supplementary Sections A.4, C.2 and C.3). That said, often in machine learning, once something begins to work (even with clear flaws), in a few short years with scale (e.g. of compute and data), better core models, and better techniques, the capabilities of a system become surprising, and can exceed human performance levels: consider the arc of progress in a short time span in generated images (GANs to DALL·E-3) and text (GPT-1 to GPT-4.5). In assessing the impact of a technology, it is thus important to keep in mind its likely future trajectory.”

These additions give readers both qualitative and quantitative insight into when and how the system fails.

3. Editor's comment: *Further details needed about screening. You mention the number of ideas generated in the first step, of which 3 are chosen by researchers. But then a number of papers are generated from random seeds. How many in that step?*

Response: We have clarified the paper screening process and added specific counts in the Supplementary Information:

- **New Supplementary Table A5** (line 302 in Supplementary Information) summarises the number of generated ideas, experimental attempts, and final manuscript write-ups at each stage.
- **Lines 144–153 (main text):** “The template-free version of The AI Scientist was readily adapted to this setting by simply prompting with the workshop’s broad theme (which was investigating deep learning limitations including where prior ideas to improve it had not worked). The overall process was then run to generate ideas, experiments, and papers, with some manual filtering at each stage (Supplementary Section A.4). This process resulted in three complete manuscripts being selected for submission. The selection was based on three criteria: whether the idea was aligned with the workshop topic, whether the code correctly implemented the proposed idea and ran without errors, and the correctness of the manuscript formatting (Supplementary Section A.4).”

This ensures transparency in the selection pipeline and clarifies the relationship between random seeds, idea screening, and final submissions.

We trust that these revisions address the Editor’s requests and improve the clarity, scope, and completeness of our manuscript.

Sincerely,
The AI Scientist Team

Dear Referees,

We thank all referees for their time and valuable feedback. We are delighted they found our work "highly promising," holding "great potential to accelerate scientific discoveries and transform the way research is conducted in the future" (Referee #5). We appreciate the recognition that our system is "novel" (Referee #2), "the first to consider [a] fully automated agent for scientific research discovery" (Referee #2), and serves as an "interesting status report" on the state of the art in this domain (Referee #3). The referees also highlighted that the paper is "technically quite solid" (Referee #1), the "methodology and use of statistics seems sound" (Referee #3), and that our open-source codebase is an "excellent resource for the community to build upon the 'AI Scientist' paradigm" (Referee #6). Finally, we are encouraged by the sentiment that this is "good and interesting work" that "should be published" (Referee #1) to spark a "necessary discussion" (Referee #1) in the community.

Below we provide a point-by-point response to each comment by the referees (our responses are in **bold**).

We have made substantial changes and believe the paper is now significantly improved thanks to your feedback. Of course, please let us know if you would like any additional changes. We believe the paper is much stronger now and we hope you agree that the paper will serve as a valuable and timely contribution to the *Nature* community.

Best regards,
The AI Scientist Team

Shortcuts to the responses:

- Referee #1
- Referee #2
- Referee #3
- Referee #4
- Referee #5
- Referee #6

Point-by-point Response

Response to Referee #1 (Remarks to the Author):

This is good and interesting work with some serious framing issues.

early work rather than rigorously peer-reviewed outlets. Thus, it is not reasonable to claim that the paper underwent the full scientific peer-review process.

We agree that our original phrasing may have sounded overstated and have now substantially toned it down in the revised manuscript, including moving away from the claim of fully automating science. We agree workshops have a much lower bar, but they are still competitive and do reject a good proportion of papers (e.g., around 30%). Our intention is only to report that the submission underwent a competitive peer-review process (one that rejects a substantial portion of human-authored papers) and that our AI-generated paper placed above roughly 55% of all human submissions. The revised text now explicitly clarifies this distinction while preserving the central point: the system demonstrated the ability to produce a paper of publishable quality under community-based peer evaluation. To make the reader fully informed, we clearly and repeatedly highlight the difference between workshop and conference papers. Specifically, the following sentences are in the paper for this purpose, which we think makes things crystal clear for the reader:

1. "In computer science, such top-tier conferences are the primary and most prestigious venues for archival rigorously peer-reviewed publication. They also have workshops with a lower, but still a non-trivial bar for peer-reviewed acceptances." (main text)
2. "One manuscript achieved high enough scores to exceed the average human acceptance threshold at a workshop, marking the first instance of a fully AI-generated paper successfully navigating a peer review process, albeit one with a lower bar." (main text)
3. "The team concluded that while one of the papers did meet the bar for workshop papers, none met the higher bar for a main ICLR conference publication." (main text)
4. "While The AI Scientist generated a peer-reviewed workshop paper, there is room for improvement to match the best human-produced science. Only one of three submissions was accepted, and workshops have much higher acceptance rates than main conferences (e.g., 70% for the ICLR 2025 ICBINB workshop [39] vs. 32% for the ICLR 2025 main conference [40]). Therefore, The AI Scientist does not yet meet the standards for top-tier publications, nor even consistently for workshops." (main text)
5. "Overall, the paper was considered a borderline workshop accept, acknowledging its valuable insights while noting the main idea is not well-motivated for a main conference." (Supplementary Section C.2)
6. "While the template-free system successfully generated a peer-reviewed workshop paper, this achievement must be contextualized. Acceptance occurred at a workshop, where papers generally report exploratory work and acceptance rates (60-80%) are much higher than at main conferences (20-30%). With only one of three submissions accepted, the system does not yet consistently meet even

In summary, I would say that, while interesting, the paper does not convincingly show that automated AI research is at a level where it can produce interesting research. The paper that was supposedly accepted was very far from an interesting discovery. And the contribution of this submission, namely the design of an agentic workflow to conduct automated research, was interesting but not quite enough to merit publication in Nature. I also think the authors could have done a better job at critically assessing the actual scientific contribution made but the system.

We appreciate hearing your perspective. While we acknowledge that the current system may not yet produce discoveries at the level of major human-led breakthroughs, as we wrote above, both the community and multiple other reviewers think this work represents an important milestone worthy of publishing in Nature. We have improved the manuscript significantly in response to your comments and those of the other reviewers. We hope you will agree that the paper is improved and now ready for publication in Nature.

Response to Referee #4 (Remarks to the Author):

Summary:

- The authors propose an AI system that autonomously performs all steps in scientific research from conception all the way to publication.
- The first phase is idea generation: the agent creates an archive of high-level research ideas and selects promising directions. It then filters ideas by checking Semantic Scholar to ensure they are not too similar to existing publications.
- The next phase is experiment execution, using two approaches: template-based (starting from existing code) and template-free (writing a starter script and iteratively improving it). After each experiment, the agent records notes in an experimental journal.
- The third phase produces a formal LaTeX write-up, which is reviewed by an Automated Reviewer that outputs numerical scores and an accept/reject decision. The reviewer was evaluated using past submissions to a machine-learning conference.
- Authors submit 3 papers generated by their agent (with guidance provided by authors) to a workshop at ICLR. 1 out of these 3 passed the threshold for acceptance.

Comments:

Results seem premature:

- The authors are exploring an important direction; their early approaches show promise and are worth pursuing further. However, at this stage, the submitted paper, particularly in making the strong claim of fully automating science, is not suitable for publication in Nature.
- Line 66: "marking the first instance of a fully AI-generated paper successfully navigating a peer review." In the ML community, workshops are primarily venues for disseminating and discussing

Thank you for taking the time to read our paper. We are happy to hear you think the work is important, well-executed, and worthy of publishing in Nature. We are sorry we did not get the tone right in our original submission. As you will see below, we have made substantial changes to address that issue. We hope to have addressed all your concerns.

To begin with, the title is unreasonably overselling the paper. It reads more like clickbait than like the title of a scientific paper. The second sentence of the abstract echoes this, by claiming that AI can automate the whole scientific process. The idea that these systems could "radically accelerate the pace of scientific discovery, ushering forward an era that could see the solution to many of society's greatest challenges" also seems premature at the very least. It is a very long way from producing a convincing simulacrum of a scientific paper to actually accelerating the pace of scientific discovery; in fact, one might argue that the generation of even more mediocre papers could slow science down, rather than the opposite. Saying that the system described in this paper represents "full end-to-end automation of science" seems about as accurate as saying that Midjourney represents the end-to-end automation of art. (For the avoidance of any doubt, this means: not very accurate at all, in fact entirely inaccurate.)

What the paper actually shows is that there exists a system which can produce a mediocre workshop paper with limited human intervention (human screening of generated ideas). This is very impressive in itself, and this impressive result deserves to be discussed on its own merits. The fact that an LLM-based agentic system can now produce a basic scientific paper, for better or worse, is important to highlight and should spark a necessary discussion about the scientific ecosystem and the nature of science. In this context, the current bombastic language rather serves to detract from the actual contribution. I think Nature should eventually publish this paper, but only after these claims have been toned down considerably.

Another way of putting this is that I, the reviewer, am convinced that the work is good and important and should be published. Well done, authors, you have succeeded. Now please stop sounding like a used car salesman or second-rate politician, and rewrite the appropriate parts of the paper in the manner of scientists talking to other scientists. Be open about what the system can and cannot do, which capabilities are likely to develop fast with better models, and which capabilities are likely to be harder to improve. Also, no need to hide (in the supplementary material) that humans curated the research ideas or that many of the ideas were quite similar to each other.

We changed the title to make it more modest, and to use the specific language requested by the editor and other reviewers ("end-to-end" vs. fully automated). The new proposed title is "Towards End-to-end Automation of AI Science with AI", which replaces the original "Entirely Automating AI Science with AI." The "towards" signifies that we have not completed that journey.

We substantially edited the summary paragraph to tone down rhetoric. Specifically, we eliminated the text "ushering forward an era that could see the solution to many of society's greatest challenges." We also made many other changes to the summary,

including eliminating mentioning things like “few dreamed [we] could automate the entire scientific process” and eliminating the use of the word “creative” to describe The AI Scientist. We have also made changes throughout the main text to reflect this more modest tone. Additionally, we expanded the section on limitations of the current system and its downsides in the Supplementary Section C.3.

To address the potential issue you raised of AI-generated content slowing science down, we added the beginning clause to the following sentence of the summary: “As with any impactful new technology, there could be significant risks, including taxing overwhelmed review systems and adding noise to scientific literature. However, if developed responsibly, such autonomous systems could greatly accelerate scientific discovery.” The paper also includes a discussion of many other potential downsides at the end of the main text, to which we added a mention of this issue as well: “The ability to automate paper generation raises significant ethical and societal concerns, including the potential to overwhelm the peer review process, artificially inflate research credentials, repurpose the ideas of others without giving proper credit, eliminate scientist jobs, and/or conduct unethical or dangerous experiments (Supplementary Section C.3).”

Regarding the claim about “the full end-to-end automation of science,” we have modified that sentence to make clear it was referring specifically to the scientific process described in the prior sentence (idea conception through producing a single paper), rather than the automation of all science.

The manuscript includes this text in the main text to let readers know that some steps had human selection (but only selection): “The overall process was then run to generate ideas, experiments, and papers. We manually filtered the most promising outputs at each stage (Supplementary Section A.4). Had this filtering not occurred, the papers under analysis would still have been produced in their final form, just along with other papers and thus at a greater total cost.”

Regarding describing “which capabilities are likely to develop fast with better models, and which capabilities are likely to be harder to improve”, we have added the following text:

“Crucially, this trajectory is not just about better models, but about the complexity of tasks AI systems can execute. Recent work suggests the length of tasks AI can reliably complete doubles every seven months [41], indicating that many current implementation and debugging bottlenecks may be resolved in the near term. However, some AI weaknesses have proved surprisingly difficult to solve, such as AI being easily fooled [42, 43] and otherwise overconfidently wrong (i.e., hallucinations [44]), though progress has been made [45, 46]. Such challenges could persist, preventing us from reliably trusting the outputs of systems like The AI Scientist. It is also not clear to what extent AI systems can produce extremely novel, creative ideas that resemble great conceptual

We did include our own assessment of the paper, which can be found in Supplementary Section C.1, C.2, and D.2. We think much of the paper makes sense, but agree some key aspects (like the motivation for the main method) were not well-explained. Then again, many human scientific papers also make such mistakes. Indeed, this paper outperformed 55% of human-authored papers in peer review. While there is noise in the process of course, the fact that the paper is good enough to be within the distribution wherein noise matters and can lead to being ranked so much higher than human-authored papers is itself a milestone (akin to an AI Scientist Turing Test). We are clear in our writeup that we do think the workshop paper could be much clearer, and that the main motivation is unclear. But as we say in the paper, the systems will only get better with time, and it is an important moment in history that we can already produce something of this quality.

This brings me to my final point. Are we ready to unleash AI scientists into the world? The authors are well aware of the dangers of flooding the internet with bad science and they do reflect on it. However, the availability of these tools will undoubtedly have the effect that many young researchers whose careers depend on papers in top tier conferences will give this a spin. In that sense the paper does us a service to let us know where we stand and what to expect.

We agree with all of these points, including that one contribution of the manuscript is announcing that the technology is now capable of producing manuscripts that could flood the system. We have expanded our discussion of these issues in the manuscript to highlight them even more than we did in the original manuscript. Specifically, we modified the following sentence in the main text to mention this issue:

“The ability to automate paper generation raises significant ethical and societal concerns, including the potential to overwhelm the peer review process, artificially inflate research credentials, repurpose the ideas of others without giving proper credit, eliminate scientist jobs, and/or conduct unethical or dangerous experiments (Supplementary Section C.3)”

The methodology and use of statistics seems sound. The paper could be clearer in the sense that a more concrete example of what the process delivered in every phase would be welcome. Also I think there is too much emphasis in the main paper in the artificial reviewer which I think is not the main achievement. More details on the workflow would have been better.

Thank you. We have addressed all reviewer’s comments regarding clarity by adding text about the paper generation pipeline, having the effect of reducing the emphasis on the reviewer. Regarding adding workflow examples, we have added a new section to the Supplementary Section A.2.8 to provide example outputs from each phase of the experiment pipeline. For the idea generation phase, we provided example outputs in the Supplementary Section A.2.7 and C.4. For the writeup phase, the deliverables are the papers themselves; sample outputs are available in our GitHub repository.

viewed over 6 million times, and open-source repositories received 13 thousand stars on GitHub. The community thus seems to find this work very interesting. Multiple other reviewers also believe it is worth publishing in Nature, including to spark a discussion in the community of how best to adapt to this technology now that it exists.

I was less impressed with the actual result. I find the bar of passing a workshop review process rather low. It would have been a lot stronger if their would have been a genuinely interesting discovery or innovation, but this is not reported.

We have toned down the claims in the paper and eliminated the use of the word creative to describe the system. However, the main milestone we report on is an AI system that can complete the entire process of producing a paper, starting with conceiving of the idea through writing and reviewing a completed manuscript. We believe that alone is a major milestone and breakthrough, even if the degree of breakthrough in the AI-generated paper itself is not large. As you note, many human science teams also produce incremental papers, and that they are still counted as helpful contributions to science. Even a few years ago, it would have been considered science fiction for AI to complete the process of coming up with an idea through to finishing writing a paper and peer reviewing it.

Of course, it would be even better if the system produced major breakthroughs. We have seen signs of it being creative, including coming up with ideas that are celebrated by the machine learning community. We have added the following example to the Supplementary Section (C.3):

“Conversely, we also observed a case where The AI Scientist generated an idea that was later pursued by a human team that was celebrated for their work. While anecdotal, this provides an example of The AI Scientist producing an idea that the community found creative and worthy of scientific exploration.

The AI Scientist-generated proposal can be found among the released papers in our open-source repository (see Code Availability). The paper by the human team [43] was published in a peer-reviewed journal (Physica D: Nonlinear Phenomena). While the core ideas were very similar, the human study executed the idea more effectively than The AI Scientist.”

In fact, I read the actual paper and it did not make a whole lot of sense to me. The sentences read like an ML paper but is it more than a correct sounding paper? I can see how this gets accepted into a workshop with the title “I can't believe it's not better”, because it didn't work really well. But besides a low bar for papers that report on research that didn't end up working very well, I actually don't think what was proposed made much sense. It would have been nice if the authors would have given their own assessment of the written paper.

leaps in science. Studying and improving AI systems on these fronts are key areas for future research.”

We additionally added the following text in Supplementary Section C.3:

“Many of the failures listed above are symptomatic of the limitations of current-generation models. As shown in Figure 1B, paper quality directly correlates with model improvement. We expect this trend to continue, based on past experience: The ability for AI to reliably complete complex, long-horizon tasks is doubling every seven months [37]. This suggests that problems related to implementation, multi-step debugging, and maintaining logical consistency throughout a long research process are likely to be solved in the near future.”

Overall, we have toned down claims considerably, as you requested. Many of those changes are listed above or in the updated Supplementary Section C.3, and there are other smaller changes woven throughout. We hope you agree we have struck a better tone. We appreciate your saying the paper should be published with this modified tone to spark a necessary discussion.

I think it should be clarified in the main paper text that the generated paper that passed peer review had a negative result. There is nothing wrong with that - negative results are important - but I think it should be pointed out.

We have added the following text: “Notably, the accepted manuscript reported a negative result, aligning with the workshop’s focus on interesting negative results.”

The section C.3. limitations and broader impact, is interesting for what it addresses as well as for what it doesn't address. It is welcome that the issues with overwhelming the review system with automatically generated papers are pointed out. This could be explored in more depth, as I think the issue of “clogging the pipes” of scientific peer review is pressing already as it is.

See above where we address the issue of “slowing science down.”

But there is a further problem. Imagine that some future AI scientist actually becomes as good, or better, as a good human researcher at ideating and executing experiments. What, then, becomes of human scientists? The authors of the paper under consideration are AI researchers; presumably they do not want to become replaced by their own creations. Or do they? What effects would this have on trust in science and in human expertise, incentives for learning, and the social fabric? I think this is an important enough issue to be discussed further in the supplemental material and also at least mentioned in the main paper text.

We agree these are important issues for society to grapple with, although they are of course not unique to science given the advances in AI and potential widespread economic disruption that looms. We do not pretend to have the answers for these

important questions, but, to start the conversation, we are happy to flag them as an important potential societal implication of this technology.

We have expanded our discussion of these issues in the manuscript to highlight them even more than we did in the original manuscript. Specifically, we modified the following sentence in the main text to mention the issue of scientist jobs potentially being eliminated:

“The ability to automate paper generation raises significant ethical and societal concerns, including the potential to overwhelm the peer review process, artificially inflate research credentials, repurpose the ideas of others without giving proper credit, eliminate scientist jobs, and/or conduct unethical or dangerous experiments (Supplementary Section C.3).”

We have expanded the Supplementary Section C.3 section titled “Limitations and Broader Impact” to address the issues you raised (and those raised by other reviewers).

Technically, the paper is quite solid. I can't find any major shortcomings, and the supplemental material is as far as I can see complete.

We deeply appreciate the time you put into the review. Thank you for noting that the paper is technically solid and should be published once we have improved the tone. We have tried to satisfy your requests, and we hope you agree the paper is now ready for publication.

Response to Referee #2 (Remarks to the Author):

Thank you for the time you spent reviewing our work. We are glad to hear you find our system “novel”, “technically sound”, and recognize that we are one of the “first to consider fully automated agent for scientific research discovery”. Please see below where we respond to each of your comments.

Summary

This paper describes a novel AI agent system for tackling the scientific research discovery process: find research question, develop research idea, run experiment to validate the idea and iterate; the final outcome of the process is to present a research paper.

To mimic the crucial peer review process in science, the authors designed an automated review system to provide feedback following machine learning conference guidelines.

The overall framework includes two settings: one called template-based AI scientist where human scientists initiated the system by providing some codes and experiment setups and the other called template-free AI scientist where the discovery is done without the initialization. One of the generated papers was accepted to a workshop in the ICLR conference (one of the major machine learning conferences) through the blind review process.

is hard to judge their actual implementation and scientific contributions. While both systems are interesting examples of human-AI collaboration, The AI Scientist demonstrates a more complete automation of the scientific process itself, which is the central advance of our work.

Nevertheless, both works are related and we have added a mention of them (and the above differences) to the Supplementary Section A.4. Thank you again for mentioning them.

We believe we have fixed all the issues you raised, and hope you agree with us that the paper is now ready for publication. Thank you for your feedback, which made the paper significantly stronger.

Response to Referee #3 (Remarks to the Author):

The papers reports on an impementation of a automated process to perform ML research using LLM agents. The process involves formulating a hypothesis, generating, debugging code to perform computer experiments, finetuning hyperparameters, generating plots, assessing visualizations, performing ablation studies and writing a paper in Latex which is auto-reviewed (and meta reviewed) by an ensemble of LLMs.

3 papers were produced and submitted to a ICLR conference workshop. It was found that one of them was above the acceptance bar, justifying the claim that this could be considered the first fully automated publishable scientific paper.

The main contribution seems to be the design and implementation of a workflow of various kinds of LLMs from OpenAI and Anthropic, that can autonomously conduct research. These LLMs need to coordinate with each other and with internet resources and datasets on HuggingFace, providing them with the correct context information in their prompt to perform their research.

I find the paper not very original in the sense that this is a rather obvious question to try to answer. The approach to achieving this result is however somewhat novel.

I found it interesting to see what the state of the art is in fully automated science, and in this sense I think this is an interesting status report.

Thank you for noting that our work is novel, and an interesting status report on the state of the art of AI Scientist technology. We believe the work is extremely interesting because it reports a new, historically important advance in technological capability related to science. There is evidence that many others think this is impactful: to date, preprints we shared with the community have received 685 citations, posts about it were

are sometimes workshop level. There are thus many different types of qualitative and quantitative evidence that support the main conclusion of the paper.

Relevant work

Two related works (later than AI Scientist-V1, but concurrent to AI Scientist-V2) also claimed generating papers accepted in ICLR 2025 workshops: Carl <https://www.autoscience.ai/blog/ai-research-automation> and Zochi <https://www.intology.ai/blog/zochi-tech-report>. Zochi also later announced another paper first-time passed the peer review of ACL main conference

Thank you for raising these. These works are related, but involve significant levels of human involvement in the generation process of the paper manuscript (e.g. LaTeX edits, conceptual figure creation) and thus do not reach the same level of automation as The AI Scientist.

Regarding Carl: First, their papers were not entirely written by AI. They admit that humans needed to write the “related works section”, “polish the final language”, and “manually edit the citations and LaTeX formatting in the papers” to match the style guide. In contrast, the AI-generated papers we submitted were entirely generated end-to-end by AI, without any modifications from humans.

Second, their papers were accepted to a track designed to have lower requirements than most standard workshop tracks at ICLR (the Tiny Papers track). According to their technical report, even the partially AI-generated papers they submitted received, on average, a rejection recommendation from reviewers. Despite that, the final decisions made by the Tiny Papers track organizers, whose aim is to be inclusive, was to accept the works (see the quite negative reviews they published for their Paper 1 and Paper 2). In contrast, our AI-generated paper received acceptance scores of 6, 7, 6, which are well above the threshold for acceptance.

Regarding Zochi/intology, their process involves humans actively steering the research, providing feedback “a few times during experimentation to help steer away from unproductive approaches.” The final manuscript also requires substantial human labor, including “creating publication-quality figures and diagrams, performing final edits to ensure papers fit within conference page limits, checking and adding citations... and properly formatting complex tables.”

In contrast to Carl and Zochi, The AI Scientist represents a greater degree of end-to-end automation. The entire workflow for each paper—from the initial idea to experimentation to the final compiled PDF—is completed autonomously without any human modification of the final paper output.

Both of their systems are not openly available and, to the best of our knowledge, the submissions were done without organizer approval nor proper IRB approval. Therefore, it

Comments

First of all, this is one of the first work that considers fully automated AI system in an “end-to-end” manner: from idea generation to peer review and finally present research outcome in a research paper, and leads to peer-reviewed publications. I reviewed the method section, and the proposed methodology is technically sound. My main concerns are on the experiment side and if the results support the claim and future promise.

Questions

Q1: This paper mainly presents one paper generated by the agent system accepted for an ICLR workshop related to challenges and limitations in applied deep learning. I noticed the scope narrowed down suggested by the editors, but I'm still concerned if one case study would represent machine learning research entirely—as noted by the authors, the workshop received only around 40 submissions. The paper could benefit from more study on different category of machine learning work.

While our paper does highlight one example paper that was accepted via peer review at an ICLR workshop, it also studies many other papers generated by the system, and on a variety of different topics within ML. Specifically, across two versions of The AI Scientist, we include the detailed analysis of, and full text of, 4 AI-generated papers in the supplementary material. Additionally, we include the full text of 50 generated papers in the open-source repository (see Code Availability). These papers span various machine learning topics such as diffusion models, grokking, language modeling, uncertainty calibration, and compositional regularization. We thus believe it does represent a systematic (not one-paper) study of many different categories of ML.

Q2: Even if a paper passes conference peer review, the claim about automating discovery could be made more cautious. For example, even from human scientists, there are also a significant amount of incremental work to improve some certain technical parts or putting two architectures together to create a new one. These types of borderline papers can be accepted by chance. It would be good to see a comprehensive analysis of the types of research AI agent would generate.

We agree that many scientific papers make a small, incremental contribution. However, many small steps can add up to a long journey resulting in a large cumulative advance. Additionally, since many, if not most, human papers are of that incremental type, we believe it is still a milestone to automatically generate that type of science. We agree that the current version of The AI Scientist cannot make major conceptual leaps or produce breakthroughs that rival some of the major leaps in human science. We have added a discussion of these issues to the Supplementary Section (C.3, Sections Paradigm-Shifting Creativity and Strategic Scientific Judgment), and mention this issue in the main text in the following sentence:

"Therefore, The AI Scientist does not yet meet the standards for top-tier publications, nor even consistently for workshops," and "it is also not clear to what extent AI systems can produce extremely novel, creative ideas that resemble great conceptual leaps in science. Studying and improving AI systems on these fronts are key areas for future research."

We also do an extensive qualitative analysis of the type of work that The AI Scientist produced (see Supplementary Section C.1 and C.2). Additionally, we made the overall tone much more cautious throughout the title and paper, as you requested, including eliminating multiple uses of the word "creative" when describing the system.

Q3: *The authors justify the future promise of the framework by demonstrating stronger models with more computational resources could improve the performance measured by the automated reviewer. As noted by the authors, workshops often have around 70% acceptance rate (way higher than 25-30% acceptance rate in main conference). In addition, the paper is significantly shorter: 4 vs 8-9 pages. The reviewers often evaluate workshop papers in a much lighter way compared to conference submissions. How do the authors justify the gap will be bridged by the improved compute or model performance? Stronger evidence would be extremely helpful for this claim to be made to such broad audience in Nature.*

We agree there is a substantial gap between the acceptance threshold for workshop and conference papers. But the recent history of ML shows that each year or two provides substantial improvements in performance. GPT-1 made a large number of mistakes, yet GPT-4 outperforms most humans on exams given to graduate and pre-graduate students (e.g. the LSAT, the GRE, etc.). GPT-1 could barely reason, but o3 reasons much, much better. In addition to these arguments being in the paper, to support our claim that this gap will narrow, we have added a summary of independent evidence from METR along with a citation to that study where the reader can learn more (See the Limitations section in the main text):

"In assessing the impact of a technology, it is thus important to keep in mind its likely future trajectory. Crucially, this trajectory is not just about better models, but about the complexity of tasks AI systems can execute. Recent work suggests the length of tasks AI can reliably complete doubles every seven months [41], indicating that many current implementation and debugging bottlenecks may be resolved in the near term."

That quote is from the main paper and the issue is also discussed in Supplementary Section C.3.

Q4: *Can you clarify how exactly human is involved in the template-based setting? Does human also formulate the problem? E.g. in your first example, before the agent system proposed the idea, do you start by giving the code for some diffusion models and prompting it to improve them? I assume yes as it is unlikely to provide code/experiment without formulating the problem.*

In the template-based setting, humans provide runnable baseline code (e.g., a simple diffusion model training script) and the following prompt: "Come up with the next impactful and creative idea for research experiments and directions you can feasibly investigate with the code provided. Note that you will not have access to any additional resources or datasets. Make sure any idea is not overfit to the specific training dataset or model, and has wider significance." All subsequent ideation is done by the system. The specific training scripts were (and remain) available in the open-source repository (see Code Availability), but we have also added them to the Supplementary Section A.1.2.

Q5: *For the generated papers accepted in the workshop, it is from the template-free approach. Did you also submit the papers generated by the template-based approach?*

Yes, the papers we submitted to the workshop were generated using the template-free approach. We did not submit any papers produced with the template-based approach to a workshop (though we do include examples of them in the supplementary material of this paper). This is stated in the paper in Supplementary Section A.5 and C.1.

Q6: *For the evaluation of the automated review system, the NeurIPS experiments spotted the issues of human reviewers as two batch of reviewers could lead to opposite decisions. Showing the overall accuracy in agreeing with the decision of one group of reviewers is great, but do you spot some patterns that can categorize LLM generated reviews?*

We have not noticed any specific tendencies or systematic biases/patterns in the LLM generated reviews, though this does not mean that they do not exist. In Figure 1C, we tested whether there is memorization of paper reviews potentially included in the training corpus of the frontier LLMs. We found that the AI reviewer could generalize to papers reviewed after the training data cutoff. That being said, there may be different undesired biases distilled from human review data used to train the underlying LLMs.

Overall, this work is the first to consider fully automated agent for scientific research discovery, but I'm mostly concerned about the limited quantitative analysis and the strong enough evidence to support the claim of "entirely automating" as a message this paper would send to the community.

We have substantially toned down the overall rhetoric, including specifically the point of "entirely automating." That includes the title and the summary, both of which were modified to make the claims more modest. We have read many of the papers and include our analyses of these papers in the main text and supplementary info. Our conclusion is that the system is capable of producing scientific work (and writeups) on its own, although its work is at best workshop level. On top of that, there is the evidence that one of the papers outperformed 55% of human-authored workshop submissions and was accepted via peer review. Finally, the quantitative evidence from our automated reviewer, which we validated is as good as humans at inter-reviewer reliability, suggest the papers